# NRF2 modulates WNT signaling pathway to enhance photodynamic therapy resistance in oral leukoplakia

Tiantian Xu[1,4], Liang Zhong [ID][2,4], Qianxi Liu[1,4], Fei Wang[1,4], Wenjing Kuang [ID][1], Jiaqi Liang[1], Dan Yang[1], Xikun Zhou [ID][3], Hongxia Dan[1], Hang Zhao[1], Taiwen Li [ID][1✉], Xin Zeng [ID][1✉], Jing Li [ID][1✉] & Qianming Chen[1,2]

## Abstract

Oral leukoplakia (OLK) is a common potentially malignant oral disorder with high risk of malignant transformation. While photodynamic therapy (PDT) offers a minimally invasive treatment for OLK, some patients show resistance to PDT and the mechanisms remain unclear. This study aims to identify key regulatory pathways driving PDT resistance in OLK. Single-cell RNA sequencing of OLK samples (three PDT-sensitive, three PDT-resistant) revealed significant NRF2 upregulation in resistant tissues. Validation across two independent cohorts ($n = 117$) confirmed that p-NRF2 levels were significantly elevated in PDT-resistant cases, exhibiting strong predictive power for treatment response (AUC > 0.8). Mechanistically, NRF2 promotes *CTNNB1* transcription, activates WNT signaling, modulates reactive oxygen species responses, and regulates keratinization, collectively contributing to PDT resistance. In a 4NQO-induced OLK mouse model, NRF2 inhibition combined with PDT effectively reversed OLK lesions and restored mucosal histology. These findings establish p-NRF2 as a valuable biomarker for guiding PDT regimens in OLK patients, reveal NRF2's role in mediating PDT resistance via the WNT signaling pathway, and highlight NRF2 inhibition as a promising strategy to enhance PDT efficacy.

**Keywords** Oral Leukoplakia; Photodynamic Therapy; Resistance; NRF2; WNT

**Subject Categories** Digestive System; Skin

## Introduction

Oral leukoplakia (OLK) is the most prevalent oral potentially malignant disorder (OPMD) in clinical practice (Kerr and Lodi, 2021). Its prevalence is reported to be 0.5% to 3.46%, and its malignant transformation rate is 1% to 9.7% worldwide (Schepman et al, 1998; Warnakulasuriya, 2020; Wils et al, 2023). The risk of malignant transformation in OLK patients is 40.8 times that of the general population (Chaturvedi et al, 2020). Traditional treatment methods include drug treatment, laser treatment, surgical resection, etc. (Hanna et al, 2024). These treatment methods, whether applied individually or in combination, have several limitations, including limited efficacy, relatively high recurrence rates, resistance development, a risk of postoperative scarring, potential impairment of oral function, and, in some cases, negative effects on facial esthetics and overall quality of life (Kerr and Lodi, 2021). Notably, similar challenges persist in the management of other OPMDs. For instance, while local treatments such as corticosteroids and other pharmacological agents can effectively alleviate symptoms in oral lichen planus (OLP) (Polizzi et al, 2023; Polizzi et al, 2024) and oral submucous fibrosis (OSF) (Tang et al, 2025), their efficacy in achieving long-term disease control and mitigating the risk of malignant transformation remains significantly limited (Lodi et al, 2012; Sun et al, 2019). The limitations of these treatments not only increase the physical and psychological burden on patients but may also lead to significant increases in socioeconomic costs. Therefore, developing treatment strategies with high efficacy and minimal side effects is crucial for improving the prognosis and quality of life of OLK patients and more effectively preventing cancer in its precancerous state (Warnakulasuriya, 2020).

Photodynamic therapy (PDT), as a non-invasive treatment method, has shown a unique role in the treatment of OLK and other OPMDs (Li et al, 2020). PDT employs specific light wavelengths to activate photosensitizers, which generate cytotoxic reactive oxygen species (ROS). This process ultimately results in the death of malignant or tumor cells that have selectively absorbed the photosensitizers, thereby reducing or eliminating the lesion (Cramer et al, 2022; Li et al, 2020). PDT is one of the most promising treatments for OLK (Chen et al, 2019). It is suitable for all types of OLK, especially those with heterogeneous OLK, moderate-to-severe epithelial dysplasia, and high expression of Ki67 in lesion tissues. Studies have shown that PDT not only holds significant promise for OLK treatment but also provides

[1]State Key Laboratory of Oral Diseases & National Center for Stomatology & National Clinical Research Center for Oral Diseases & Research Unit of Oral Carcinogenesis and Management & Chinese Academy of Medical Sciences, West China Hospital of Stomatology, Sichuan University, Chengdu 610041 Sichuan, China. [2]Stomatology Hospital, School of Stomatology, Zhejiang University School of Medicine, Zhejiang Provincial Clinical Research Center of Oral Diseases, Key Laboratory of Oral Biomedical Research of Zhejiang Province, Cancer Center of Zhejiang University, Hangzhou, China. [3]State Key Laboratory of Biotherapy and Cancer Center, West China Hospital, Sichuan University and Collaborative Innovation Center for Biotherapy, Chengdu 610041, P. R. China. [4]These authors contributed equally: Tiantian Xu, Liang Zhong, Qianxi Liu, Fei Wang. ✉E-mail: litaiwen@scu.edu.cn; zengxin@scu.edu.cn; lijing1984@scu.edu.cn

therapeutic benefits in managing premalignant lesions in other OPMDs such as OLP (Binnal et al, 2022). Notably, PDT has been reported to effectively alleviate ulcerative symptoms in refractory erosive OLP (Rakesh et al, 2018) and improve mouth-opening capacity in OSF patients (Wan et al, 2023). However, despite its potential in OPMDs management, the therapeutic efficacy of PDT remains suboptimal, with only 60% of OLK patients responding favorably (Jing et al, 2024; Li et al, 2019), while a substantial proportion exhibit resistance. This underscores the necessity for a deeper investigation into the underlying mechanisms of PDT resistance to improve treatment outcomes. Research on PDT resistance mechanisms has primarily focused on tumors, especially non-oral tumors (Xie et al, 2021; Zhang et al, 2022), with few studies investigating disease-specific resistance mechanisms for OLK (Yang et al, 2024). Therefore, to reduce the malignant transformation rate of OLK patients and improve their survival and quality of life, clinical oncology and precision medicine must investigate the mechanism of PDT resistance in OLK patients, search for biomarkers that predict PDT efficacy, and establish an accurate prognostic assessment system.

Despite the increasing application of PDT in the management of OPMDs, its clinical translation remains constrained by an incomplete understanding of the molecular mechanisms underlying resistance and the lack of validated biomarkers for predicting treatment response and facilitating patient stratification. To address these limitations, this study systematically investigates the cellular and molecular determinants of PDT resistance in OLK through a comprehensive, multi-dimensional approach. Here, we used single-cell RNA-sequencing (scRNA-seq) to identify cell types and markers that contribute to PDT resistance in OLK. We demonstrated that epithelial cells are the key cell subsets that resist PDT treatment and identified NRF2, particularly its phosphorylated form, p-NRF2, as a reliable predictor of PDT response in OLK. Mechanistically, NRF2 binds to functional antioxidant response element (ARE) sites in the promoter region of CTNNB1, driving the transcriptional activation of β-CATENIN and subsequently activating the WNT signaling pathway. NRF2 promotes PDT resistance by modulating the WNT signaling pathway, reshaping cellular antioxidant responses, and regulating keratinization processes in OLK cells. Furthermore, we found that Brusatol, an NRF2 inhibitor, significantly enhances the efficacy of PDT in treating OLK.

# Results

## Single-cell transcriptomic analysis suggests that epithelial cells are the main contributing cells to OLK PDT resistance

To investigate the predominant cellular populations and their molecular mechanisms contributing to PDT resistance, lesion tissues from nine patients with clinical manifestations of OLK were collected and subjected to scRNA-seq following a predefined workflow. According to the predefined inclusion and exclusion criteria, six OLK patients further underwent standardized PDT treatment and were grouped based on their responses to sequential PDT treatments (Figs. 1A,B and EV1A). We conducted rigorous quality control of the scRNA-seq data by assessing key cellular feature metrics. The number of detected genes per cell (nFeature_RNA) predominantly ranged

between 500 and 8000, and the total RNA counts (nCount_RNA) displayed a normal distribution without evidence of abnormally high expression levels. Additionally, the proportion of mitochondrial gene content (percent_mt) was below 20% in the vast majority of cells, indicating minimal contamination by low-quality or apoptotic cells. Collectively, these parameters demonstrate that the dataset meets the quality thresholds necessary for reliable downstream analyses (Fig. EV1B–D). Overall, we captured the transcriptomes of 14 major cell types based on known cell type-specific gene features (Figs. 1C,D and EV1E,F). The cell clustering derived from PDT-resistant tissues and PDT-sensitive tissues exhibited obvious differences, as exemplified by epithelial cells, T cells, myeloid cells and fibroblasts (Fig. 1E). Interestingly, DEGs expression analysis revealed that the epithelial cell population exhibited the most pronounced differential gene expression profile, with 473 genes upregulated and 358 genes downregulated (Fig. 1F). Next, we further re-clustered the epithelial cells and categorized them into basal, cycling, and differentiated subpopulations based on marker genes characteristic of oral stratified squamous epithelial cells. These marker genes correspond to the well-established functional roles of cells within the epidermis (Figs. 1G and EV1G). Next, we performed differential gene expression comparisons and GO analysis between the sensitive and resistant groups for the three epithelial cell subtypes. We found that the epithelial cell subtype in the resistant group was enriched in terms such as antioxidant stress response, epithelial keratinization barrier, and epithelial adhesion (Fig. 1H). Overall, these results suggest that PDT resistance in OLK may be primarily related to the functional properties of its epithelial cells, such as antioxidant and epithelial barrier capacity.

## Activation of NRF2 in tissues is associated with PDT-resistance in OLK patients

To elucidate the cell-intrinsic transcriptional mechanisms underlying PDT resistance in epithelial cells, we performed SCENIC analysis to predict TFs activities and identify potential regulators of differentially expressed genes. Of note, NRF2 (NFE2L2), a pivotal TF involved in antioxidant response pathways, displayed significantly enhanced activity across all epithelial subgroups across PDT resistant tissue when compared to the sensitive tissue (Fig. 2A,B). Pathway enrichment analysis of NRF2 target genes highlighted its role in augmenting antioxidant stress response and reinforcing epithelial barrier integrity during PDT resistance (Figs. 2C and EV2A; Appendix Table S2). Further investigation into feature scores related to these pathways in OLK epithelial cells from both PDT-resistant and sensitive groups revealed that NRF2-associated antioxidant response element feature scores and keratinization scores were notably higher in the epithelial cells of the PDT-resistant group than in the sensitive group (Fig. 2D).

IHC analysis demonstrated the upregulation of NRF2 and p-NRF2 expression in the tissues of PDT-resistant OLK patients from the discovery cohort (Figs. 2E,F and EV2B). As p-NRF2 represents the transcriptionally active form of NRF2, future studies will concentrate on its function. In a larger internal validation cohort, we confirmed the increased p-NRF2 expression in PDT-resistant OLK tissues compared to PDT-sensitive samples (Fig. 2G). Moreover, logistic regression analysis revealed that elevated p-NRF2 levels were significantly associated with a poorer response to PDT treatment, suggesting that p-NRF2 can serve as an independent prognostic factor for PDT efficacy. In contrast, PDT

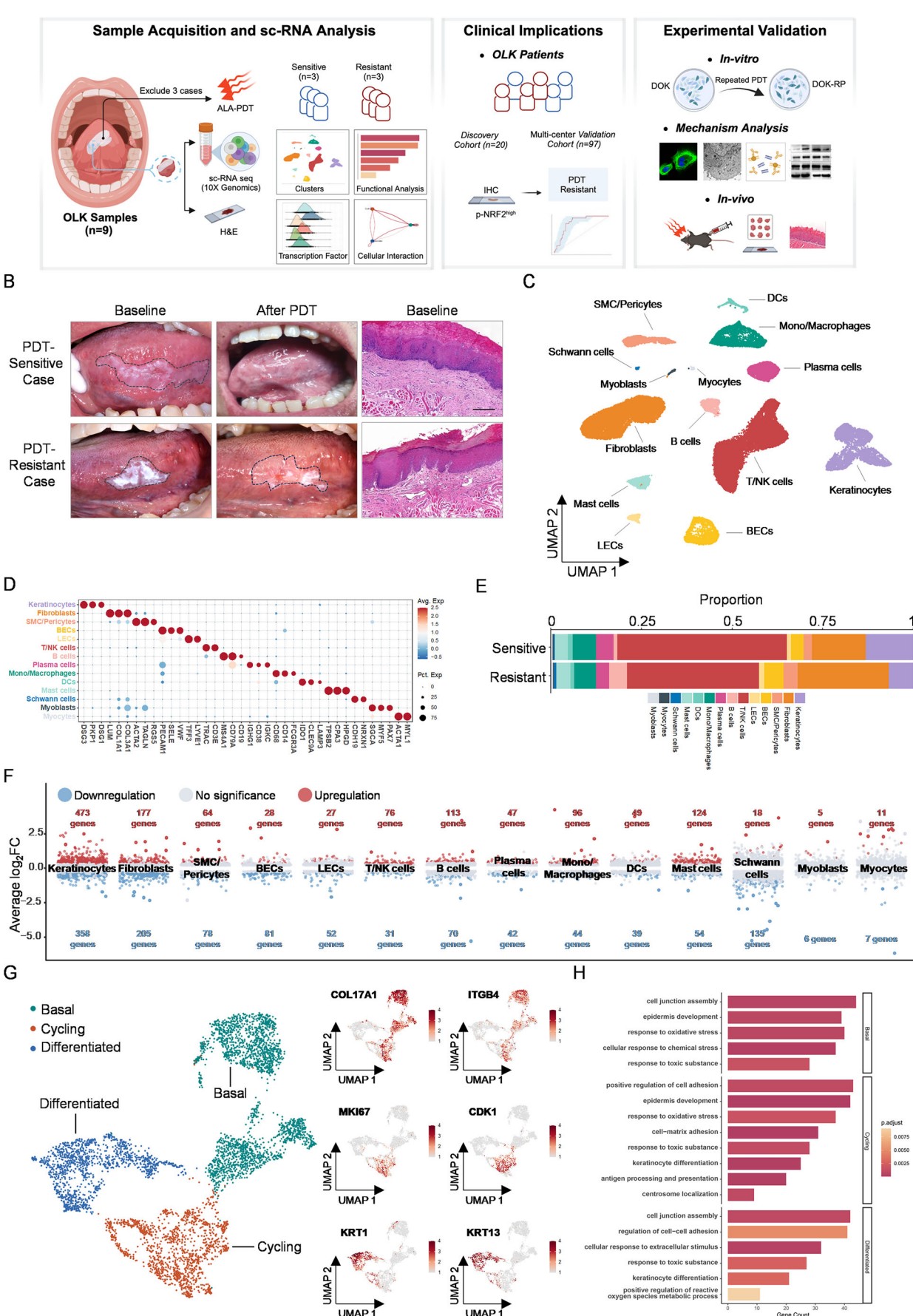

**Figure 1. Single-cell transcriptome analysis emphasizes the importance of epithelial cells in OLK PDT resistance.**

(A) Schematic illustration of the overall experimental design used in this study. Lesion tissues from nine OLK patients were subjected to scRNA-seq; three were excluded based on predefined criteria. The remaining six received standardized ALA-PDT and were classified as PDT-sensitive ($n = 3$) or -resistant ($n = 3$) based on clinical response. scRNA-seq data were analyzed for cell clustering, functional annotation, transcription factor activity, and cell–cell interactions. p-NRF2 expression was assessed by IHC in a discovery cohort ($n = 20$) and a multicenter validation cohort ($n = 97$), confirming its predictive value for PDT resistance. A PDT-resistant DOK cell line (DOK-RP) and an OLK mouse model were used to evaluate the therapeutic potential of NRF2 inhibition combined with PDT. (B) Representative clinical and pathological images of OLK tissues exhibiting sensitivity or resistance to PDT treatment. Scale bar: 200 μm. (C) UMAP visualization of single cells from six OLK patients, revealing 14 distinct clusters. (D) Average expression levels of canonical marker genes across the 14 clusters. Statistical significance was assessed using the Wilcoxon rank-sum test. (E) Average cellular composition of OLK samples from PDT-resistant and -sensitive patients. (F) Number of differentially expressed genes (DEGs) in each cell cluster between PDT-sensitive and -resistant groups, identified using the Wilcoxon rank-sum test. (G) UMAP visualization of epithelial cell subclusters with representative marker genes annotated for each cluster. (H) GO analysis of enriched pathways in three epithelial cell subtypes with DEGs. *P* values were calculated using a hypergeometric test (Fisher's exact test) and adjusted for multiple comparisons using the Benjamini–Hochberg method. Data information: Source data are available online for this figure.

efficacy showed no significant correlation with epithelial dysplasia or demographic characteristics such as age and sex (Table 1). Receiver operating characteristic (ROC) curve analysis suggested that p-NRF2 could effectively predict PDT efficacy, with an area under the curve (AUC) value of 0.825 (Fig. 2H). By analyzing p-NRF2 expression levels in the external validation cohort and applying the model established from the internal validation cohort to generate ROC curves, we demonstrated that p-NRF2 serves as a prognostic biomarker for PDT efficacy, consistently exhibiting superior performance across different clinical centers (Fig. 2I,J). Given the close association between NRF2 nuclear localization and its transcriptional activity, we further examined the nuclear levels of p-NRF2 across three cohorts. The results demonstrated a significant upregulation of nuclear p-NRF2 expression in the resistant groups of the multi-center cohorts (Fig. EV2C). Correlation analysis further revealed a positive correlation between p-NRF2 expression and the degree of epithelial keratinization (Figs. 2K and EV2D).

## NRF2 is transiently activated in a self-constructed PDT-resistant OLK cell model

To visualize the disease landscape of PDT resistance in OLK patients in vitro, we established a PDT-resistant OLK cell model named DOK-RP cells for subsequent studies (Fig. 3A,B). After exposure to PDT, DOK cells exhibited various stages of apoptosis within two to four hours, characterized by chromatin condensation, cell shrinkage, and mitochondrial changes, as observed via TEM. On the other hand, DOK-RP cells exhibited no apoptotic features yet displayed elevated mitochondrial electron density and a rounded morphology, indicating a resistant response to PDT-induced stress (Figs. 3C and EV3A).

Notably, compared to the parental cells, DOK-RP cells exhibited lower ROS levels (Figs. 3D and EV3B) and reduced intracellular MDA accumulation following PDT treatment (Fig. 3E). Furthermore, the GSH/GSSG ratio was significantly lower in DOK cells (Fig. 3F). Moreover, compared to the parental cells, the expression levels of SLC7A11 and HMOX1, the key regulators of oxidative stress response, were significantly upregulated in DOK-RP cells after PDT treatment (Figs. 3G and EV3C). These observations imply that DOK-RP cells minimize intracellular ROS levels after PDT by activating the anti-oxidative stress system.

Given previous evidence linking cellular keratinization to PDT resistance, cellular IF analysis revealed a denser and more robust overall keratin filament network in DOK-RP cells (Fig. 3H). Markers indicative of squamous epithelial keratinization, such as

KRT1, KRT10, and LOR, were upregulated in DOK-RP cells (Fig. 3I). This increased keratinization might explain the reduced uptake of photosensitizers observed in PDT-resistant cells due to their enhanced epithelial barrier function (Fig. EV3D). Protein levels of p-NRF2 were found to be consistently elevated at multiple time points following PDT treatment in DOK-RP cells compared to DOK cells (Figs. 3J and EV3E). Additionally, p-NRF2 exhibited more pronounced nuclear localization in DOK-RP cells (Fig. 3K). Additional IF experiments further confirmed higher p-NRF2 expression in DOK-RP cells (Fig. EV3F). These findings collectively demonstrate that NRF2 is activated in both in vivo tissues and in vitro cell models of PDT resistance.

## Activation of NRF2 induces resistance to PDT in the OLK cells

To detect whether aberrant NRF2 expression contributes to increased resistance of OLK cells to PDT, we constructed a DOK cell line stably overexpressing NRF2 (Figs. 4A and EV4A,B). The survival rates of DOK cells with NRF2 overexpression were significantly higher across various PDT power intensities compared to the control group (Fig. 4B). Consistently, treatment with Brusatol, an inhibitor of NRF2, led to a decrease in intracellular p-NRF2 protein levels and promoted PDT-induced cell death in DOK cells, effectively reversing PDT resistance in DOK-RP cells (Figs. 4C–E and EV4C).

Subsequently, we sought to determine if NRF2's enhanced regulation of the PDT-resistant phenotype in cellular models operates through a shared mechanism involving antioxidant stress response and cellular keratinization, as suggested by scRNA-seq data. Flow cytometry analysis revealed that NRF2 overexpression reduced PDT-induced intracellular ROS levels, a response similar to that observed in DOK-RP cells. Treatment with Brusatol led to a further increase in PDT-induced cytotoxic ROS in both DOK and DOK-RP cells (Fig. 4F). Similarly, the glutathione metabolic system followed this trend, with NRF2-overexpressing cells showing an increased GSH/GSSG ratio after PDT treatment, which was subsequently decreased upon Brusatol administration (Fig. 4G).

We also found that the expression of intracellular antioxidant stress-related markers and cellular keratinization program markers was upregulated by NRF2 overexpression, which was reversed by adding the NRF2 inhibitor Brusatol (Fig. 4H,I). These findings suggest that the NRF2 inhibitor rescues the resistant phenotype of DOK-RP cells by inhibiting NRF2's dual roles in anti-oxidative stress and keratinization promotion, ultimately allowing epithelial cells to withstand PDT-induced cell death.

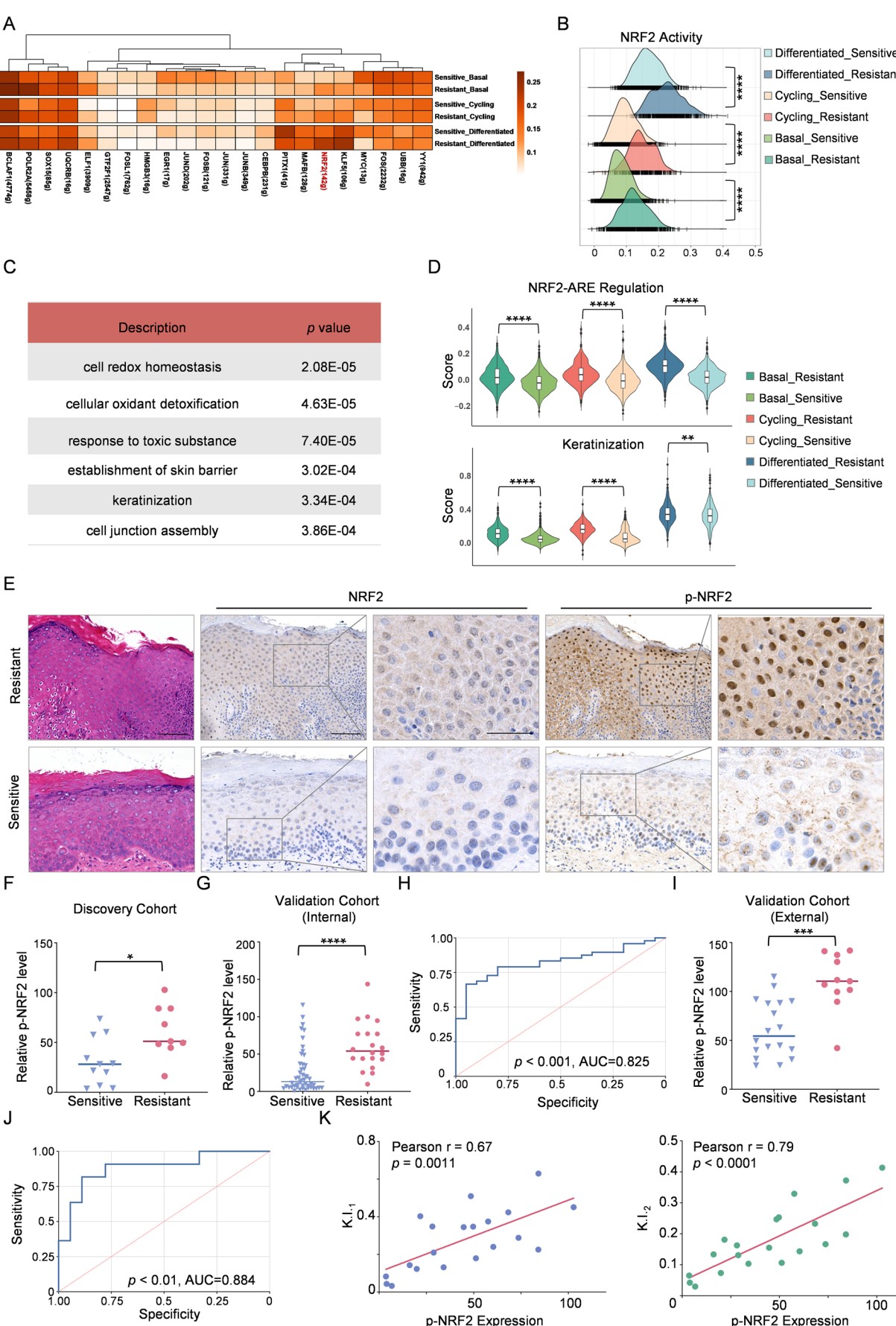

**Figure 2. Increased NRF2 expression and activation in OLK patient tissues correlate with reduced PDT responsiveness.**

(A) SCENIC analysis reveals differential transcription factor activity between PDT-resistant and sensitive OLK samples. (B) Ridge plots showing NRF2 activity across epithelial subpopulations in PDT-sensitive and -resistant OLK samples. Statistical significance was assessed using the Wilcoxon rank-sum test. (C) Representative pathway enrichment results for NRF2 downstream regulated target genes. $P$ values were calculated using a hypergeometric test (Fisher's exact test) and adjusted for multiple comparisons using the Benjamini–Hochberg method. (D) Violin plots depicting the activity of the NRF2-ARE and keratinization pathways across different epithelial cell subpopulations in PDT-sensitive and -resistant OLK samples. Subpopulations include Basal (sensitive: $n = 1466$ cells; resistant: $n = 799$), Cycling (sensitive: $n = 1042$; resistant: $n = 330$), and Differentiated (sensitive: $n = 820$; resistant: $n = 474$). Each data point represents an individual cell. The center line indicates the median; boxes represent the interquartile range (25th–75th percentiles); whiskers extend to 1.5× the interquartile range; dots represent individual data points. Wilcoxon rank-sum test was used for statistical comparison. (E) Representative IHC staining images for NRF2 and p-NRF2 in PDT-sensitive and -resistant OLK samples from the discovery cohort. Scale bar: 200 μm. (F) Relative expression of p-NRF2 in PDT-sensitive ($n = 11$) and resistant ($n = 9$) OLK samples from the discovery cohort in IHC analysis (Student's t-test). (G) Relative expression of p-NRF2 in PDT-sensitive ($n = 48$) and resistant ($n = 20$) OLK samples from the internal validation cohort in IHC analysis (Student's t-test). (H) ROC curve analysis of p-NRF2 for predicting PDT efficacy in OLK patients from the internal validation cohort ($n = 68$). Group comparison was assessed using the Wilcoxon rank-sum test; AUC significance was evaluated by DeLong's test. (I) Relative expression of p-NRF2 in PDT-sensitive ($n = 18$) and resistant ($n = 11$) OLK samples from the external validation cohort in IHC analysis (Student's t-test). (J) ROC curve analysis of p-NRF2 for predicting PDT efficacy in OLK patients from the external validation cohort ($n = 29$). Group comparison was assessed using the Wilcoxon rank-sum test; AUC significance was evaluated by DeLong's test. (K) Correlation analysis of p-NRF2 expression and tissue K.I. in the discovery cohort. Statistical significance for the Pearson correlation coefficient was assessed using Student's t-test ($n = 20$). Data information: In (F, G, I, K), each data point represents an individual patient (biological replicate). Significance is indicated as $*p < 0.05$, $**p < 0.01$, $***p < 0.001$, $****p < 0.0001$. Exact $p$-values for these comparisons are provided in Appendix Table S1. Source data are available online for this figure.

**Table 1. Correlation of PDT effectiveness with clinicopathologic factors in the internal validation cohort.**

| Clinicopathological feature | Total (*n*) | Odds radio | *p*-Value |
|---|---|---|---|
| Age (continuous) | 68 | 0.991 | 0.751 |
| Gender (female vs. male) | 68 | 0.984 | 0.981 |
| Degree of dysplasia (positive vs. negative) | 68 | 0.897 | 0.881 |
| p-NRF2 expression (continuous) | 68 | 1.032 | 0.002 |
| Degree of keratinization (high vs. low) | 68 | 1.621 | 0.463 |

## WNT signaling pathway is involved in OLK's PDT resistance

Next, we utilized the CellChat algorithm to evaluate the impact of differentially expressed ligand-receptor pairs in OLK tissues. Despite a lower percentage of epithelial cells in the PDT-resistant group, our analysis revealed stronger intercellular communication among cells in the resistant group compared to the sensitive group (Fig. EV5A–C). Heatmap showed that the cell communication signal intensity of WNT, NOTCH and EPHB were increased in all epithelial cell subpopulations of the PDT-resistant group (Fig. EV5D). By analyzing the communication strength of WNT signaling pathway-associated ligand and receptor pairs, we identified key components such as WNT ligands and Frizzled receptors that play essential roles in cellular communication (Fig. EV5E). The feature scores for the WNT signaling pathway were significantly higher in the PDT-resistant group compared to the sensitive group (Fig. 5A).

Furthermore, our findings showed that the expression of β-CATENIN, a key component of the WNT signaling pathway, was upregulated in PDT-resistant tissues compared to sensitive tissues in our OLK cohort (Fig. 5B). There was a positive correlation between β-CATENIN expression and the degree of epithelial keratinization (Fig. EV5F). In DOK-RP cells, we observed increased intracellular β-CATENIN accumulation and higher levels of p-GSK3β. Additionally, several ligands of the WNT signaling pathway, including WNT3A, WNT4, and WNT10A, were

upregulated in DOK-RP cells, and that their corresponding receptor, FZD6, was also highly expressed in DOK-RP cells (Fig. 5C). These results support the involvement of the WNT signaling pathway in OLK's resistance to PDT.

## NRF2 regulates PDT resistance by promoting *CTNNB1* gene transcription, thereby activating the Wnt signaling pathway

Evidence suggests a possible interaction and regulation between NRF2 and the WNT signaling pathway (Fragoulis et al, 2022), leading us to hypothesize that NRF2 may contribute to PDT resistance in OLK patients by activating the WNT pathway. Consistent with this hypothesis, we observed that NRF2 overexpression upregulated WNT signaling components, including ligands (WNT3A, WNT4, and WNT10A), the receptor (FZD6), and key pathway elements (β-CATENIN and p-GSK3β) in DOK cells, whereas NRF2 inhibition with Brusatol had the opposite effect (Fig. 5D). Importantly, we found that treatment with the WNT signaling pathway agonist BML284 increased markers of oxidative stress response (SLC7A11 and HMOX1) and cellular keratinization programs (KRT10, KRT1, and LOR). Conversely, the WNT signaling pathway inhibitor IWR-1 suppressed these effects (Fig. 5E).

Moreover, WNT agonists partially reversed the increased intracellular ROS levels caused by NRF2 downregulation during PDT treatment, and partially restored treatment sensitivity in cells where NRF2 was inhibited (Figs. 5F and EV5G). This reversal effect was observed in both DOK cells and DOK-RP cells (Figs. 5G and EV5H). On the other hand, in cells with NRF2 overexpression, WNT inhibitors partially increased the intracellular ROS levels induced by PDT, thereby partially reversing the PDT resistance induced by NRF2 overexpression in DOK cells (Figs. 5H and EV5I). Additionally, previous studies have reported that the promoter region of the gene encoding β-CATENIN (*CTNNB1*), a key effector of the WNT signaling pathway, contains an antioxidant response element (ARE) sequence that can be bound by NRF2 (Fig. 5I). To validate this, we simulated the predicted ARE sequence of *CTNNB1* using AlphaFold3 and employed the HDOCK method to predict potential binding interactions between NRF2 and the *CTNNB1*-ARE sequence. The results indicated a high likelihood of

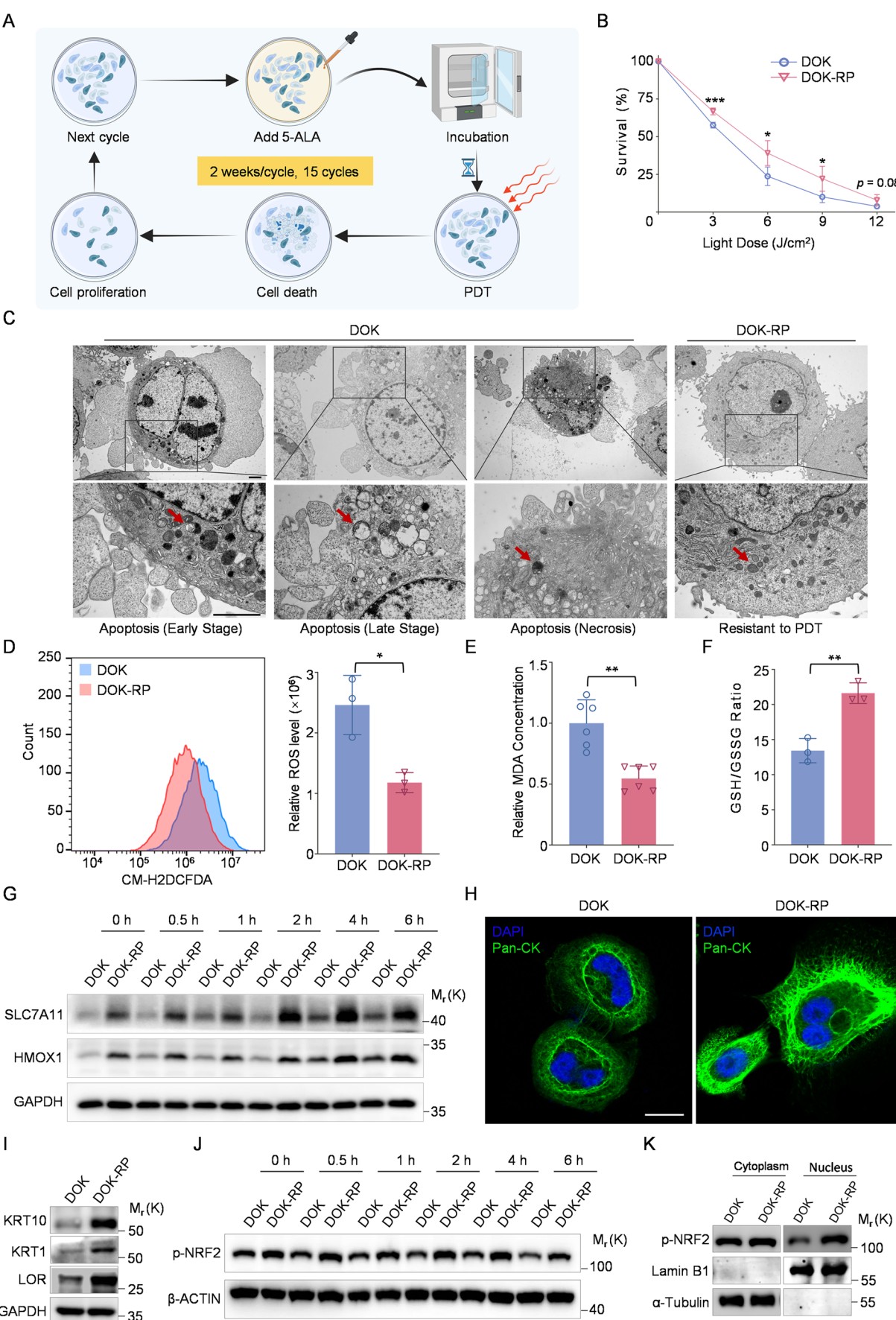

**Figure 3.  NRF2 and p-NRF2 are increased in the constructed PDT-resistant human OLK cell line model.**

(A) Process snapshot of the establishment of PDT-resistant human OLK cell line model. (B) Cytotoxicity assays were used to evaluate differences in the killing ability of PDT on DOK and DOK-RP cells at various light doses. Data represent biological replicates ($n = 4$). Statistical significance was assessed using Student's t-test. (C) Representative transmission electron microscopy images illustrating subcellular structural changes in DOK and DOK-RP cells treated with PDT. The red arrows indicate mitochondrial structures in the TEM image. Scale bar: 2 μm. (D) ROS levels in PDT-treated DOK and DOK-RP cells analyzed by flow cytometry (left), with corresponding quantification (right). Data represent biological replicates ($n = 3$). Statistical comparison was performed using Student's t-test. (E) MDA levels in PDT-treated DOK and DOK-RP cells (Student's t-test). Data represent biological replicates ($n = 6$). (F) Ratio of GSH to GSSG levels in DOK and DOK-RP cells treated with PDT (Student's t-test). Data represent biological replicates ($n = 3$). (G) Dynamic expression of SLC7A11 and HMOX1 protein in DOK and DOK-RP cells following PDT treatment across multiple time points. (H) Representative images illustrating the morphology and intensity of the cytokeratin network in DOK and DOK-RP cells using immunofluorescence. Scale bar: 20 μm. (I) Immunoblotting analysis of the expression of KRT10, KRT1, LOR in DOK and DOK-RP cells. (J) Immunoblotting analysis of the expression of p-NRF2 in DOK and DOK-RP cells following PDT treatment across multiple time points. (K) Immunoblotting analysis of the expression of p-NRF2 in the nucleus and cytoplasm of DOK and DOK-RP cells. Data information: In (B, D, E, F), data are presented as mean ± SD. Each data point represents an individual biological replicate. Significance is indicated as *$p < 0.05$, **$p < 0.01$, ***$p < 0.001$. Exact p-values for these comparisons are provided in Appendix Table S1. Source data are available online for this figure.

interaction between the NRF2 protein and the *CTNNB1*-ARE sequence (Figs. 5J and EV5J). Subsequently, our CUT&RUN experiments confirmed significant binding of NRF2 to the ARE sequence within the *CTNNB1* promoter region in DOK cells. Notably, NRF2 enrichment in the *CTNNB1* promoter region was significantly enhanced in DOK-RP cells compared to DOK cells (Fig. 5K). Meanwhile, *CTNNB1* mRNA expression is upregulated in DOK-RP cells compared to DOK cells (Fig. 5L). Collectively, these findings suggest that NRF2 can regulate the WNT signaling pathway to induce PDT resistance, potentially by acting as a TF for *CTNNB1*.

### NRF2 inhibitor effectively enhance the efficacy of PDT for OLK

To evaluate whether NRF2 inhibitors can enhance the efficacy of PDT in OLK, we generated a 4NQO-induced OLK mouse model, which is the classic model for OLK research. Then, OLK-bearing mice were treated with NRF2 inhibitor alone, PDT alone, or NRF treatment plus PDT (Figs. 6A and EV6A,B). At the end of the treatment, we observed that the mice in the COMBO group had significantly fewer cauliflower-like or verrucous protruding lesions on the tongue compared to other groups, with some hyperplastic lesions shrinking or even disappearing. In contrast, the lesions in other groups showed varying degrees of progression and malignant transformation (Fig. 6B,C). Subsequently, we analyzed the efficacy scores and found that compared with the three control groups, the COMBO group significantly inhibited or even reversed the progression of the lesions (Fig. 6D), and the degree of epithelial dysplasia was significantly lower (Fig. 6E–G). Further analysis revealed that the OLK tissues in the combined treatment group had fewer Ki67-positive cells than other groups (Fig. 6H,I). Additionally, no significant differences in body weight or evidence of organ toxicity were observed among the four groups at the experiment's end (Fig. EV6C,D). The above evidence indicates that NRF2 inhibition combined with PDT treatment can more effectively prevent the malignant progression of OLK, and treatment with NRF2 inhibitors can be used as an alternative strategy for enhancing PDT efficacy.

## Discussion

OLK is the most prevalent type of OPMDs, with a high rate of progression to oral squamous cell carcinoma (Aguirre-Urizar et al, 2021; Chaturvedi et al, 2020). Traditional surgical treatments can

lead to maxillofacial deformities, causing significant physical and emotional distress for patients and substantial societal burden, and yet, the mortality rate is high (Radaic et al, 2023; Warnakulasuriya, 2020). Unfortunately, there is currently no universally accepted treatment approach in clinical practice, leaving many patients living with the condition indefinitely. Notably, regardless of the treatment modality, approximately 22% of OLK patients may experience recurrence (Bhattarai et al, 2024). In such cases, PDT offers a more favorable option than traditional surgical approaches due to its reduced trauma and faster recovery. These advantages not only mitigate the physical and psychological burden associated with repeated treatments but also improve the feasibility of long-term disease management (Jing et al, 2024). Although PDT is widely used for various malignancies and precancerous conditions, some OLK patients remain resistant to this treatment, and the underlying mechanism remains elusive (Peralta-Mamani et al, 2024; Wang et al, 2024). Our study demonstrates a marked elevation of NRF2 expression in PDT-resistant OLK patients compared to sensitive tissues, which is associated with poor treatment outcomes. Mechanistically, NRF2 plays a central role in mediating PDT resistance by activating the WNT signaling pathway, which in turn influences cellular antioxidant responses and keratinization processes. This discovery provides a theoretical foundation for the synergistic treatment of OLK using PDT and offers a novel strategy to enhance the therapeutic efficacy of OLK management (Fig. 7).

Currently, the response rate of PDT in the treatment of OLK patients remains unsatisfactory, with a low complete remission rate and a high recurrence rate of ~60% (Chen et al, 2019; Gaimari et al, 2016), highlighting the need for reliable biomarkers to predict treatment outcomes and guide personalized therapies. In clinical settings, there is a dearth of biomarkers capable of accurately forecasting the effectiveness of PDT in OLK patients. Here, we explored the potential of p-NRF2 as a predictor for PDT efficacy in OLK patients within a multi-center study cohort. scRNA-seq analysis revealed higher NRF2 transcriptional activity in PDT treatment-resistant OLK patients than in the sensitive group, which was confirmed by IHC experiments in independent cohorts from multiple research centers. Our findings suggest that tissue-level p-NRF2 expression can serve as a reliable indicator of OLK patients' response to PDT therapy, offering high sensitivity and accuracy. Our team previously developed a clinical model utilizing learning algorithms to predict short-term PDT efficacy for OLK patients (Wang et al, 2024). Integrating NRF2 into this model may enhance prediction accuracy and sensitivity, providing a genetic foundation for tailored treatment strategies in OLK patients.

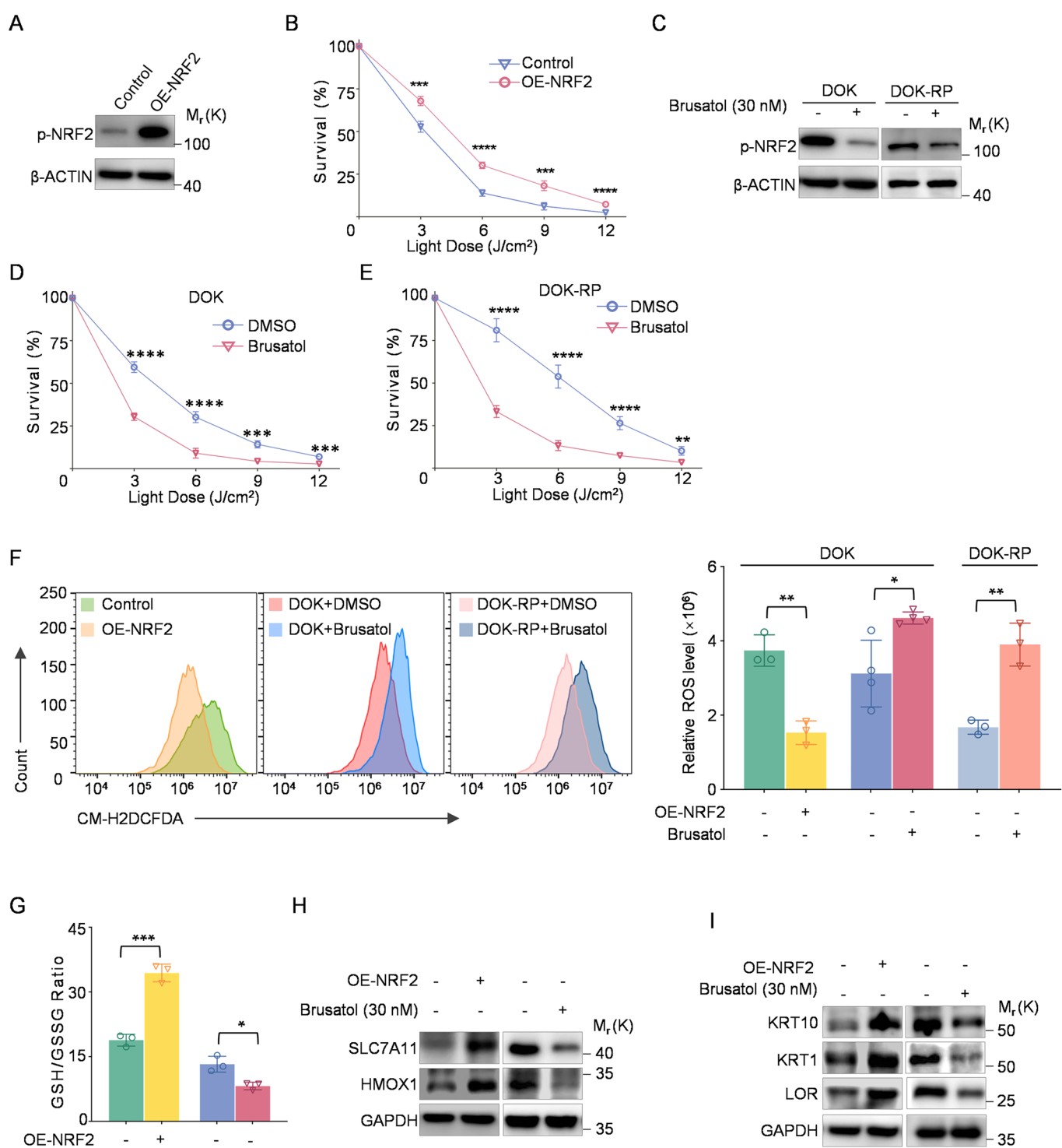

The clinical application of PDT faces several challenges. Current photosensitizers, such as 5-ALA, are primarily activated by short-wavelength lasers, resulting in low tissue penetration (Kim et al, 2020; Xie et al, 2021). The absorption of photosensitizers is impeded, and their accelerated excretion reduces their accumulation and activation efficiency in target cells (Alvarez and Sevilla, 2024). Additionally, the activation of antioxidant systems by PDT further eliminates ROS,

leading to decreased cell death (Wang et al, 2023). In the case of OLK, abnormal epithelial dysplasia and excessive keratin accumulation further impair laser penetration and photosensitizer absorption (Cai et al, 2023a; Shang et al, 2024). While research on PDT resistance mechanisms mainly focuses on tumors, little attention has been given to non-tumor conditions like OLK (Yang et al, 2024), and the absence of in vitro cell models hampers the understanding of resistance

**Figure 4. NRF2 mediates PDT resistance in OLK cell models by reducing oxidative stress and promoting cellular keratinization.**

(A) Immunoblotting analyzes the expression of p-NRF2 in NRF2 overexpressing DOK cells. (B) Cytotoxicity assays comparing the varied effectiveness of PDT in NRF2 overexpressing DOK cells versus controls. Data represent biological replicates ($n = 4$). Statistical comparison was performed using Student's t-test. (C) Immunoblotting analyzes the expression of p-NRF2 in DOK and DOK-RP cells following Brusatol treatment. (D) Cytotoxicity assays comparing the varied effectiveness of PDT in Brusatol-treated DOK cells versus controls. Data represent biological replicates ($n = 4$). Statistical comparison was performed using Student's t-test. (E) Cytotoxicity assays comparing the varied effectiveness of PDT in Brusatol-treated DOK-RP cells versus controls. Data represent biological replicates ($n = 4$). Statistical comparison was performed using Student's t-test. (F) Flow cytometry analysis of ROS levels in PDT-treated DOK cells overexpressing NRF2 (OE-NRF2, $n = 3$), Brusatol-treated DOK cells ($n = 4$), and Brusatol-treated DOK-RP cells ($n = 3$), along with their respective controls (DOK Control: $n = 3$; DMSO-treated DOK: $n = 4$; DMSO-treated DOK-RP: $n = 3$). Corresponding statistical comparisons are shown on the right. Data represent biological replicates. Statistical significance was assessed using Student's t-test. (G) Ratio of GSH to GSSG in NRF2-overexpressing DOK cells, Brusatol-treated DOK cells, and their respective controls after PDT treatment. Data represent biological replicates ($n = 3$). Statistical comparison was performed using Student's t-test. (H) Immunoblotting evaluates SLC7A11 and HMOX1 expression in NRF2-overexpressing DOK cells, Brusatol-treated DOK cells, and in their respective controls. (I) Immunoblotting evaluates KRT10, KRT1, and LOR expression in NRF2-expressing DOK cells, Brusatol-treated DOK cells, and in their respective controls. Data information: In (B, D, E, F, G), data are presented as mean ± SD. In (F, G), each data point represents an individual biological replicate. Significance is indicated as *$p < 0.05$, **$p < 0.01$, ***$p < 0.001$, ****$p < 0.0001$. Exact $p$-values for these comparisons are provided in Appendix Table S1. Source data are available online for this figure.

mechanisms (Ebrahimi et al, 2024; Zhuang et al, 2024). In this study, we created the first OLK cell model resistant to PDT treatment using a multi-round, incremental power method. Our model mimics clinical PDT resistance observed in OLK patients, characterized by reduced cytotoxic ROS levels via activated antioxidant stress pathways and decreased photosensitizer absorption through activation of epithelial keratinization. We observed elevated expression of p-NRF2 protein compared in our model, consistent with scRNA-seq data and tissue protein analyses, emphasizing NRF2's importance in OLK PDT resistance. We also found that NRF2 mediates resistance by reducing PDT-induced ROS levels and promoting keratinization. Currently, the strategies to enhance PDT efficacy primarily focus on photosensitizers, such as using nanomedicine delivery systems and developing innovative photosensitizers (Huang et al, 2022; Jiang et al, 2024). Our study shows that incorporating the NRF2 inhibitor Brusatol into the photosensitizer incubation system can enhance OLK cells' response to PDT and reverse the resistant phenotype. Targeting NRF2 as an adjuvant strategy may improve PDT efficacy in OLK patients, potentially by integrating NRF2 inhibitors into existing drug delivery systems or using them as adjuvants to photosensitizers, offering new hope for PDT-resistant OLK patients.

Activation of the WNT signaling pathway is a common driving event in human cancers, conferring cancer cells with self-renewal and growth properties and contributing to therapeutic resistance in various malignancies (Song et al, 2024; Yu et al, 2021). Our study identified heightened communication intensity of the WNT pathway in the PDT-resistant group through cell communication analysis. This pathway was activated at the single-cell transcriptional, and tissue protein levels, and in our constructed PDT-resistant OLK cell model, supporting its role in chemotherapy resistance. In other diseases, particularly liver cancer, NRF2 and the WNT pathway exhibit interplay, where WNT3A stimulates NRF2, and NRF2 either stabilizes β-CATENIN or promotes its transcription to activate the WNT pathway (Fragoulis et al, 2022; Rada et al, 2015). Our data revealed that NRF2 activates multiple WNT pathway molecules, including ligand-receptor pairs and key pathway elements, in OLK cells. Specifically, NRF2 is able to bind to the ARE sequence in the promoter region of the β-CATENIN encoded gene *CTNNB1*, acting as a TF to promote the transcription of β-CATENIN. Notably, we found that WNT pathway activation can partially restore PDT sensitivity when NRF2 is inhibited. Both pathways share downstream mechanisms related to PDT resistance,

including the antioxidant stress response and keratinization. These findings suggest that NRF2 mediates PDT resistance partly through the WNT pathway, emphasizing the need for further exploration of NRF2's specific roles in various diseases. The synergy and interaction between the NRF2 and WNT pathways in PDT resistance suggest that NRF2's function extends beyond antioxidant stress response, possibly contributing to resistance in OLK patients by activating diverse downstream networks.

Our study proposes a combined therapeutic strategy in which the application of an NRF2 inhibitor in conjunction with PDT significantly enhances PDT efficacy, effectively preventing the malignant progression of OLK to oral cancer. In PDT-resistant OLK patients, elevated NRF2 expression likely activates the WNT signaling pathway, reducing photosensitizer absorption and inhibiting the production of cytotoxic ROS during photocatalysis, leading to resistance to PDT treatment. By inhibiting NRF2, Brusatol can reverse this mechanism. The combined treatment of NRF2 inhibition and PDT maximally suppresses the progression of OLK lesions, with efficacy confirmed by both visual assessment and pathological grading. Furthermore, Ki67 staining, indicated that the lesions in the combination treatment group exhibited reduced proliferative activity, suggesting a diminished risk of metastasis and malignant invasion (Cai et al, 2023b; Jayaraman et al, 2022). Moreover, localized inflammation is widely reported in patients with OPMDs, such as OLK and OLP (Louisy et al, 2024), playing a critical role in the progression of precancerous lesions (Sun et al, 2016). Considering the critical role of NRF2 in inflammation regulation (Cuadrado et al, 2018; Lu et al, 2016), further studies are warranted to optimize its inhibition strategy and timing, with the aim of enhancing PDT responsiveness while minimizing the risk of malignant transformation. This finding provides additional evidence in support of the efficacy of the combined therapy.

In conclusion, our investigation elucidates a key mechanism underlying PDT resistance in OLK, wherein elevated NRF2 expression activates the WNT signaling pathway, suppresses cytotoxic ROS production, and promotes epithelial keratinization, ultimately impairing PDT efficacy. Moreover, p-NRF2 expression was identified as a potential predictive biomarker for PDT responsiveness, providing a basis for individualized treatment strategies. These findings offer a new perspective on the role of NRF2 in mediating PDT resistance. Furthermore, this study proposes a promising therapeutic strategy combining NRF2 inhibition with PDT to enhance treatment efficacy,

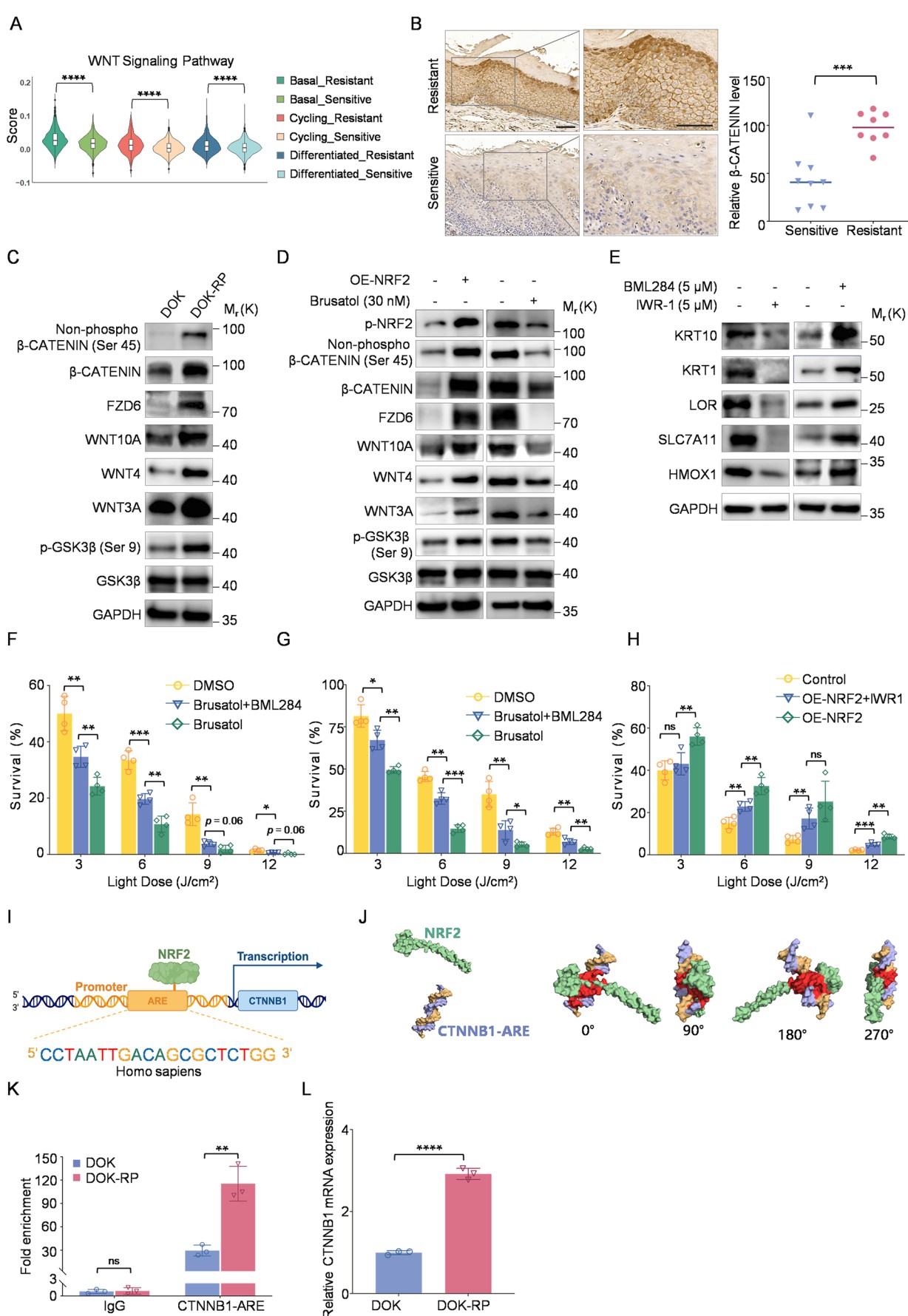

◀ **Figure 5. The communication and activity of the WNT signaling pathway in PDT-resistant OLK patients and cell models are enhanced and regulated by NRF2.**

(A) Violin plots showing WNT signaling pathway activity in different epithelial cell subpopulations in PDT-sensitive and -resistant OLK samples. Subpopulations include Basal (sensitive: $n = 1466$ cells; resistant: $n = 799$), Cycling (sensitive: $n = 1042$; resistant: $n = 330$), and Differentiated (sensitive: $n = 820$; resistant: $n = 474$). Each data point represents an individual cell. The center line indicates the median; boxes represent the interquartile range (25th–75th percentiles); whiskers extend to 1.5× the interquartile range; dots represent individual data points. Wilcoxon rank-sum test was used for statistical comparison. (B) Representative images of IHC staining for β-CATENIN in PDT-sensitive and -resistant OLK samples from the discovery cohort (left, scale bar: 200 μm). Quantification of average optical density in PDT-sensitive ($n = 9$) and -resistant ($n = 8$) OLK samples from the discovery cohort in IHC analysis (right, Student's t-test). (C) Immunoblotting analyzes the expression of WNT signaling pathway-related markers in DOK-RP and DOK cells. (D) Immunoblotting analyzes the protein expression of WNT signaling pathway-related markers in NRF2-overexpressing DOK cells, Brusatol-treated DOK cells, and their respective controls. (E) Immunoblotting analyzes the expression of KRT10, KRT1, LOR, SLC7A11, and HMOX1 in DOK cells treated IWR1 or BML284. (F) Assessing the change in response of DOK cells to PDT treatment following combined treatment with BML284 and/or Brusatol. Data represent biological replicates ($n = 4$). Statistical comparison was performed using Student's t-test. (G) Assessing the change in response of DOK-RP cells to PDT treatment following combined treatment with BML284 and/or Brusatol. Data represent biological replicates ($n = 4$). Statistical comparison was performed using Student's t-test. (H) Assessing the change in response of NRF2-overexpressing cells to PDT treatment following treatment with IWR1. Data represent biological replicates ($n = 4$). Statistical comparison was performed using Student's t-test. (I) Schematic diagram illustrating the binding of NRF2 to the ARE sequence in the promoter region of the *CTNNB1* gene. (J) The binding process between the protein (NRF2, green) and nucleic acid (ARE sequence in the *CTNNB1* promoter, purple) is displayed in surface representation, with the red regions indicating the interface area of the binding between the two molecules. (K) The CUT&RUN-qPCR assay was used to detect the binding of NRF2 to the ARE sequence in the *CTNNB1* promoter region. Data represent biological replicates ($n = 3$). Statistical comparison was performed using Student's t-test. (L) Relative mRNA expression of *CTNNB1* in DOK and DOK-RP cells. Data represent biological replicates ($n = 3$). Statistical comparison was performed using Student's t-test. Data information: In (B, F, G, H, K, L), data are presented as mean ± SD. Each data point represents an individual biological replicate. Significance is indicated as *$p < 0.05$, **$p < 0.01$, ***$p < 0.001$, ****$p < 0.0001$, ns: not significant ($p \geq 0.05$). Exact p-values for these comparisons are provided in Appendix Table S1. Source data are available online for this figure.

enabling the reversal of OLK lesions to normalized mucosa or low-grade, low-risk lesions and reducing the risk of malignant transformation into oral cancer.

# Methods

### Reagents and tools table

| Reagent/Resource | Reference or Source | Identifier or Catalog Number |
|---|---|---|
| **Experimental models** | | |
| C57BL/6 | Shanghai Model Organisms Center, Inc | |
| **Recombinant DNA** | | |
| pLV-NRF2 | VectorBuilder | VB900004-1506ejy |
| **Antibodies** | | |
| HMOX1 | Abcam | Cat #ab13243 |
| NRF2 | Abcam | Cat #ab137550 |
| NRF2 | Proteintech | Cat #16396-1-AP |
| Phospho-Ser40-NRF2 | Abcam | Cat #ab76026 |
| SCL7A11 | Abcam | Cat #ab13243 |
| Phospho-Ser9-GSK-3β | Cell Signaling | Cat #2669 |
| GSK-3β | Abcam | Cat #ab32391 |
| WNT3A | Proteintech | Cat #26744-1-AP |
| WNT4 | Huabio | Cat #RT1662 |
| WNT10A | Huabio | Cat #HA500356 |
| Cytokeratin 10 | Huabio | Cat #ST05-43 |

| Reagent/Resource | Reference or Source | Identifier or Catalog Number |
|---|---|---|
| Cytokeratin 1 | Huabio | Cat #SN72-08 |
| Loricrin | Proteintech | Cat #55439-1-AP |
| FZD6 | Cell Signaling | Cat #5158 |
| Pan-Cytokeratin | Abcam | Cat #ab7753 |
| KI67 | Abcam | Cat #ab16667 |
| GAPDH | Proteintech | Cat #10494-1-AP |
| beta-catenin | Proteintech | Cat #51067-2-AP |
| α-tubulin | Proteintech | 11224-1-AP |
| Non-phospho (Active) β-Catenin (Ser45) | CST | Cat #19807 |
| β-actin | Proteintech | Cat #20536-1-AP |
| Fuorochrome-conjugated secondary antibody | ZSGB-BIO | Cat #ZF-0311 |
| HRP-conjugated sencond antibody | Zsbio | Cat #PV-6001 |
| **Oligonucleotides and other sequence-based reagents** | | |
| PCR primers | Tsingke | Appendix Table S7 |
| **Chemicals, enzymes, and other reagents** | | |
| BML284 | Selleck | Cat #S8178 |
| Brusatol | Selleck | Cat #S7956 |
| IWR-1 | Selleck | Cat #S7086 |
| 5-aminolevulinic acid | Fudan Zhangjiang | NA |
| 4NQO | Sigma | Cat #N8141 |
| DAB Kit | DAKO | Cat #GK500705 |
| DAPI-containing ProLong Gold antifade reagent | Invitrogen | Cat #P36935 |

| Reagent/Resource | Reference or Source | Identifier or Catalog Number |
|---|---|---|
| Hydrocortisone | MedChemExpress | Cat #HY-N0583 |
| Hematoxylin | Beyotime | Cat #C0107 |
| Cell counting kit-8 | Beyotime | Cat #C0037 |
| BCA assay | Boytime | Cat #23225 |
| TRIzol | Thermo Fisher | Cat #15596018 |
| HiScript III All-in-one RT SuperMix kit | Vazyme | Cat #R333-01 |
| SYBR Green Master Mix | Thermo Fisher | Cat #4472908 |
| Rhodamine-phalloidin | Yeasen | Cat #40734ES75 |
| Chloromethyl-2′,7′-dichlorodihydrofluorescein diacetate, acetyl ester (CM-$H_2$DCFDA) | Invitrogen | Cat #C6827 |
| Lipid Peroxidation MDA Assay Kit | Abcam | Cat #ab118970 |
| GSSG/GSH Quantification Kit | Dojindo | Cat #G263 |
| Skin Dissociation Kits | Miltenyi | Cat #130-101-540 |
| Tribromoethanol | Tigergene | Cat #2402A |
| Thermo Scientific NE-PER Nuclear and Cytoplasmic Extraction Reagents Kit | Thermo Scientific | Cat #78833 |
| Hyperactive pG-MNase VAHTS ® Universal DNA Library CUT&RUN Assay Kit for PCR/qPCR | Vazyme | Cat #HD101 |
| Gel Bead & Multiplex Kit, Chromium Chip Kit, Chromium Single Cell 3′ V3 Reagent Kits | 10× Genomics | |
| **Software** | | |
| R (version 3.2.3) | | |
| GraphPad Prism (version 9.1) | | |
| SPSS (version 26) | | |
| ImageJ | | |
| **Other** | | |
| 10× Genomics Chromium Controller Instrument | 10× Genomics | |
| GentleMACS | Miltenyi | |
| Illumina NovaSeq | Illumina | |
| PDT instrument for clinical application (Human) | LG (Chongqing, China) | LG-PDT-02 |
| PDT instrument for experimental application (Cells and animals) | BHY (Tianjin, China) | BHY-PDT-TT200S |

## Clinical patients

Our study included both male and female patients, with consistent findings observed across both sexes. Key inclusion criteria were: (i) age ≥18 years and ≤80 years; (ii) diagnosis of OLK according to the 2005 WHO criteria (Warnakulasuriya et al, 2007); (iii) eligibility for PDT according to previously established indications (Chen et al, 2019; Yang

et al, 2024). Key exclusion criteria were: (i) pregnancy or breastfeeding; (ii) presence of other oral mucosal diseases, current oral cancer, or a history of oral malignancy (iii) severe systemic illness or psychiatric disorders that could interfere with treatment adherence; (iv) inability to tolerate PDT due to medical contraindications or poor general condition. Additionally, the inclusion criteria for OLK lesion tissues in the scRNA-seq cohort also take lesion size into consideration. Specifically, the lesion must be sufficient to accommodate histopathological biopsy and single-cell sample preparation while retaining enough tissue for subsequent PDT treatment. All patients underwent the standard PDT protocol as described in prior studies (Chen et al, 2019; Yang et al, 2024). All lesions received a maximum of three sessions of PDT at two-week intervals. Treatment response was evaluated 4 weeks after the last treatment. The sensitive group was also designated as patients with >20% reduction of lesions and no recurrence or aggravation at the 6-month follow-up. The resistant group was defined as patients who showed less than 20% reduction in lesion size following PDT, or those with a reduction of 20% or more who experienced relapse at the 6-month follow-up. This study included three research cohorts: the discovery and internal validation cohorts from the Chengdu center (West China Hospital of Stomatology, Sichuan University) and the external validation cohort from the Beijing center (Peking University School and Hospital of Stomatology). All patient information included in the article is presented in Appendix Tables S3–6.

## scRNA-seq analysis

OLK tissues were enzymatically dissociated using the Skin Dissociation Kit, generating a single-cell suspension with a viability exceeding 85% and a cell concentration ranging from 700 to 1200 cells/μL, meeting the requirements for scRNA-seq. scRNA-seq libraries were constructed using the 10× Genomics Chromium Controller and Chromium Single Cell 3′ V3 Reagent Kits (10× Genomics, Pleasanton, CA, USA). The libraries were sequenced on the Illumina NovaSeq platform and mapped to the human reference genome (GRCh38) using CellRanger software (version 4.0). Sequencing reads were aligned to the human reference genome (GRCh38) using CellRanger software (version 4.0). Raw gene expression matrices were subsequently processed and converted into Seurat objects using the Seurat R package (version 3.2.3), followed by integration of all samples for downstream analysis. The quality control was employed to filter out cells with <500 genes, >20% mitochondrial genes, and unique molecular identifiers <700. Harmony was used to remove the batch effect. Principal component analysis was performed on the scaled data, focusing on the top 2000 highly variable genes, with the first 10 principal components used for Uniform Manifold Approximation and Projection (UMAP) construction. Cell clustering was achieved using the "FindClusters" function with resolutions ranging from 0.2 to 1.2, depending on the cell type. Marker genes for each cell cluster were identified using the "FindAllMarkers" function with the Wilcoxon rank-sum test, applying the criteria of logFC > 0.25 and $p < 0.05$ (Campbell et al, 2023), and subsequently annotated based on established biomarkers. Differentially expressed genes (DEGs) between the sensitive and resistant groups within each cell type were determined using the "FindMarkers" function, and defined by | logFC| > 0.3 and an adjusted $p$-adjust < 0.01. To infer the potential biological functional differences in the epithelial subtypes between resistant and sensitive groups, the DEGs between groups for each

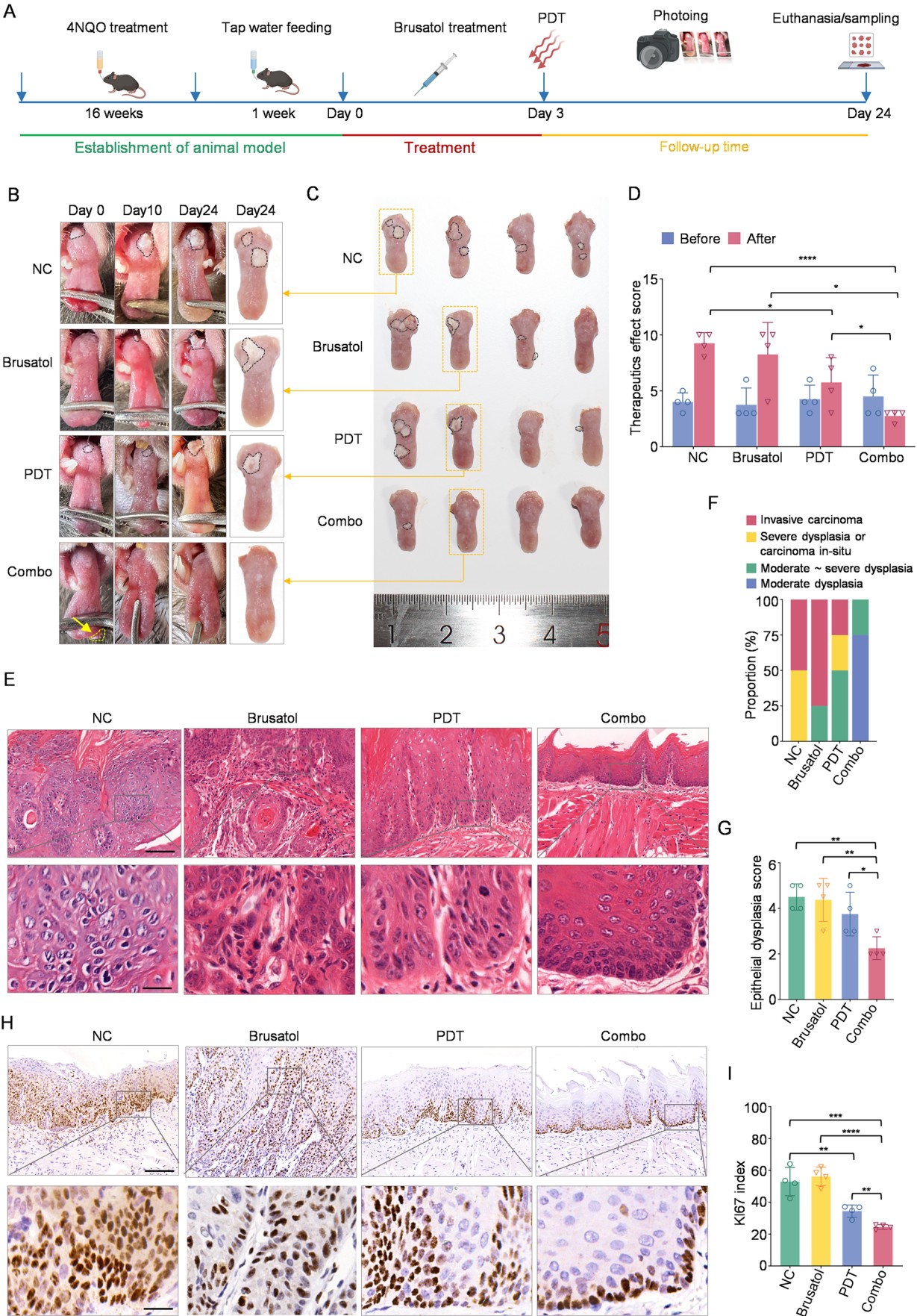

**Figure 6.   NRF2 inhibitors combined with PDT synergistically prevent the malignant progression of OLK in vivo.**

(A) Schematic timeline of mouse OLK model establishment and treatment strategy. (B) Representative images of visible tongue lesions in the control and experimental groups before and after treatment ($n = 4$ mice per group). (C) Images of tongue lesions in all experimental mice at the end of experiment ($n = 4$ mice per group). (D) Evaluation of clinical outcomes for tongue lesions in each group of mice before and after treatment using the therapeutic effect score ($n = 4$ mice per group). Statistical comparison was performed using Student's t-test. (E) Representative images of HE staining in samples from each group. Scale bar: upper, 100 μm; lower, 20 μm. (F) The proportion of invasive carcinoma, carcinoma in situ/severe dysplasia, moderate to severe dysplasia and moderate dysplasia in each group ($n = 4$ mice per group). (G) Evaluation of epithelial dysplasia in mouse OLK lesions within each group, using an epithelial dysplasia score ($n = 4$ mice per group). Statistical comparison was performed using Student's t-test. (H) Representative images of Ki67 IHC staining in samples from each group. Scale bar: upper, 100 μm; lower, 20 μm. (I) IHC analysis of the proportion of Ki67-positive cells in each group ($n = 4$ mice per group). Statistical comparison was performed using Student's t-test. Data information: In (D, G, I), data are presented as mean ± SD. Each data point represents an individual biological replicate. Significance is indicated as $*p < 0.05$, $**p < 0.01$, $***p < 0.001$, $****p < 0.0001$. Exact p-values for these comparisons are provided in Appendix Table S1. Source data are available online for this figure.

epithelial cell subtype were filtered using a threshold of |logFC| > 0.2 and p-adjust < 0.05, and then performed Gene Ontology (GO) enrichment analysis to functionally annotate these DEGs, and visualized the enriched terms that achieved a significance level of p < 0.05. The Single-Cell Regulatory Network Inference and Clustering (SCENIC) algorithm was employed to identify transcription factors (TFs) in epithelial cells within OLK. Gene expression data were integrated using CellChat to evaluate differences in hypothesized cell-cell communication modules, using the default CellChatDB as the ligand-receptor database. Cell type-specific interactions were inferred by identifying overexpressed ligands or receptors within specific cell groups, followed by the detection of enhanced ligand-receptor interactions.

## Hematoxylin and eosin staining (HE)

Following standard protocols, HE staining was performed (Xie et al, 2022). Tissue slides were photographed by a high-resolution scanner (Aperio VERSA, Leica).

## Immunohistochemical (IHC) staining

IHC staining was conducted according to previous studies (Xu et al, 2021). Tissue slides were photographed by a high-resolution scanner (Aperio VERSA, Leica). The Ki67 index is determined by randomly selecting five fields of view, each containing ~300–500 cells. The percentage of Ki67-positive nuclei is calculated for each field, and the final Ki67 index is obtained by averaging these values (Zhang et al, 2023). The immunostaining analysis was conducted by a researcher blinded to the clinicopathological information of the samples.

## Keratinization index (K.I.)

The K.I. quantifies the proportion of the stratum corneum's depth relative to the total epidermal depth and includes two sub-indices: $K.I._1$ represents the ratio of the cornified layer's depth to the total epidermis measured up to the tips of the dermal papillae. $K.I._2$ represents the ratio of the cornified layer's depth to the total epidermis measured down to the bases of the rete ridges (Brenner et al, 1982). The average value of $K.I._1$ and $K.I._2$ provides an overall measure of the stratum corneum depth.

## Cell culture and treatment

The human dysplastic oral keratinocyte cell line DOK was grown in Dulbecco's modified Eagle's medium (DMEM) with 10% fetal bovine serum (FBS) 100 U penicillin, 0.1 mg/mL streptomycin, and 400 ng/mL hydrocortisone at 37 °C with 5% $CO_2$. Based on the stimulation conditions, cells were pre-incubated with the following drugs: Brusatol (30 nM, 4 h), BML284 (5 μM, 12 h), or IWR1 (5 μM, 12 h). The final concentration of dimethyl sulfoxide (DMSO) used as a solvent did not exceed 0.1%.

## Establishment of PDT-resistant OLK cell model

Cells undergoing ALA-PDT treatment were seeded in 96-well, 24-well, or 6-well plates and were pre-incubated in FBS-free medium containing 5-ALA (0.5 mM) in the dark for 4 h. Following incubation, cells were exposed to red light at a wavelength of 635 nm (BHY-PDT-TT200S, Tianjin, China), with the laser spot size matched to the well diameter. The irradiation was performed according to the specified power settings (Yang et al, 2024), with energy fluence determined by the equation: Energy fluence ($J/cm^2$) = Optical power density ($W/cm^2$) × Exposure time (seconds). After irradiation, cells were further cultured until the designated time points for subsequent experiments. To establish a PDT-resistant OLK cell line, DOK cells were subjected to multiple rounds of PDT treatment with gradually increasing light doses (0.5–8 $J/cm^2$). After each round of treatment, surviving cells were cultured and proliferated. Following 15 rounds of PDT treatment, a stable PDT-resistant OLK cell model was successfully constructed (Fig. 3A).

## Construction of NRF2 stably overexpressing DOK cell line

Recombinant lentiviruses (pLV-NRF2 and pLV) were obtained from VectorBuilder (Shanghai, China). DOK cells were infected with the recombinant lentiviral vector carrying the NRF2 coding sequence pLV-NRF2 to facilitate stable genomic integration and sustained overexpression of NRF2. An empty vector (pLV) was used as a negative control to exclude potential phenotypic effects associated with lentiviral infection. DOK cells were infected with these vectors (MOI = 100) and screened with puromycin (5 μg/mL) for 15 days. RT-qPCR and Immunoblotting (IB) analysis confirmed NRF2 overexpression at the mRNA and protein levels. Successfully transfected cell lines were maintained in complete medium with puromycin (1 μg/mL) for further analysis.

## Cell viability assay

Cells were seeded in 96-well plates at a density of 10,000 cells per well and cultured in complete medium for 24 h. Subsequently, cells

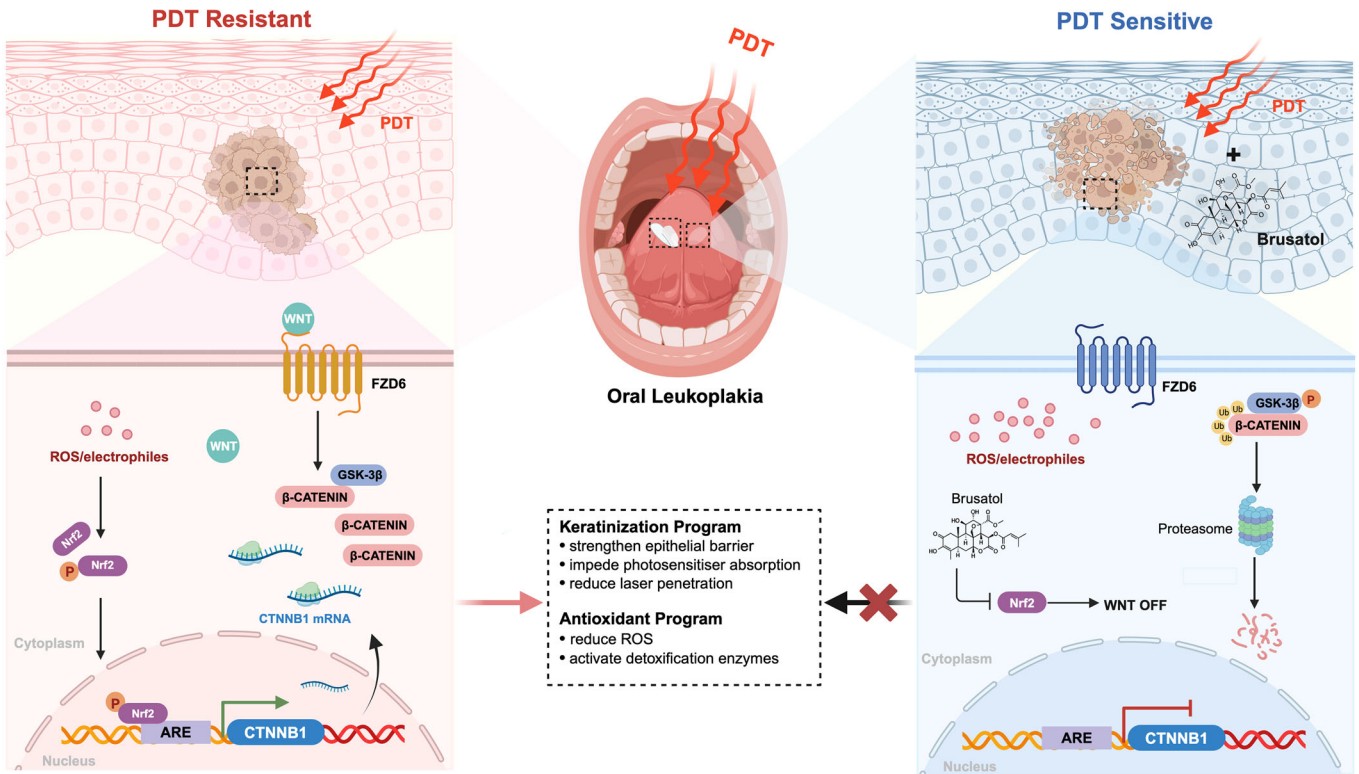

**Figure 7. Molecular mechanism of NRF2 mediates PDT treatment resistance in OLK patients and corresponding targeted sensitization strategies.**

Transitory NRF2 activation in OLK tissues upregulates the WNT signaling pathway, increasing cell keratinization and countering PDT-induced high oxidative stress, thereby reinforcing the epithelial barrier and collectively promoting PDT resistance (left panel). Nevertheless, the NRF2 inhibitor brusatol has been demonstrated to suppress aberrant NRF2 signaling, block WNT pathway transduction, inhibit cell keratinization and restore cellular sensitivity to cytotoxic ROS, thereby increasing susceptibility to PDT treatment (right panel).

were subjected to various treatments and assayed using the Cell Counting Kit-8 (CCK-8) according to the manufacturer's instructions. Each assay was performed at least in triplicate.

## Transmission electron microscopy (TEM)

The tissue was prefixed with 3% glutaraldehyde, then postfixed in 1% osmium tetroxide, dehydrated in a series of acetone solutions, infiltrated with Epon 812 for an extended period, and embedded. Semithin sections were stained with methylene blue, while ultrathin sections were cut with a diamond knife and stained with uranyl acetate and lead citrate. Sections were examined using a JEM-1400-FLASH TEM.

## Measurement of malondialdehyde (MDA) and GSH/GSSG

According to the manufacturer's instructions, MDA and GSH/GSSG levels in pretreated cell lysates were measured using MDA and GSH/GSSG assay kits, respectively.

## Measurement of reactive oxygen species

Intracellular reactive oxygen species (ROS) levels were assessed using the ROS-sensitive fluorescent probe chloromethyl-2′,7′-dichlorodihydrofluorescein diacetate, acetyl ester (CM-H$_2$DCFDA), following the manufacturer's protocol. After staining, fluorescence

signals were quantified either by fluorescence microscopy combined with ImageJ software or by measuring the mean CM-H$_2$DCFDA fluorescence using flow cytometry.

## Flow cytometry assay

Flow cytometry was conducted using the Agilent NovoCyte Advanteon. To determine the PpIX content, cells were incubated with 0.5 mM 5-ALA for 4 h, then collected and immediately analyzed with the flow cytometer. The mean fluorescence intensity (MFI) of PpIX was measured through the PerCP channel. To measure ROS levels, cells were collected following PDT treatment and immediately analyzed using the flow cytometer. The mean fluorescence intensity of the CM-H2DCFDA probe was measured in the FITC channel.

## Immunofluorescent (IF) staining

IF was conducted according to previous studies (Xie et al, 2022). The images were visualized by FLUOVIEW FV3000 (Olympus, Japan).

## RT-qPCR

Total RNA was isolated from cells using TRIzol, followed by reverse transcription. RT-qPCR was performed on the QuantStudio 3 with SYBR Green. The primers used in this experiment were shown in Appendix Table S7.

## Immunoblotting (IB) analysis

Cells were lysed using RIPA lysis buffer supplemented with protease and phosphatase inhibitors. Total protein was extracted and quantified using the BCA protein assay. Equal amounts of protein (20–30 µg) were separated by SDS-PAGE and transferred onto PVDF membranes. Membranes were blocked with 5% non-fat milk and incubated overnight at 4 °C with primary antibodies at the indicated dilutions, followed by incubation with HRP-conjugated secondary antibodies. Target proteins were visualized using enhanced chemiluminescence detection. Antibody dilutions used in this study are listed in Appendix Table S8.

## Nuclear and cytoplasmic fractionation assay

The nuclear and cytoplasmic fractionation assay was performed according to the manufacturer's instructions. The isolated nuclear and cytoplasmic fractions were analyzed using standard IB procedures.

## Protein-nucleic acid docking

The X-ray structure of NFE2L2_HUMAN (UniProt ID: Q16236) was obtained from UniProt (PDB ID: 7X5E). Protein modeling was performed using AlphaFold3 based on the *CTNNB1* sequence. Protein-nucleic acid docking was carried out using the HDOCKlite v1.1 local server (Huang and Zou, 2008; Yan et al, 2020). HDOCK is an efficient molecular docking algorithm that combines physical and bioinformatics-based methods to develop accurate scoring functions for biomolecular interactions (Mijit et al, 2023). In protein-nucleic acid docking, HDOCK samples all possible binding modes and ranks the sampled binding poses based on the scoring function. The docking results were evaluated using two metrics: (i) the docking score, where more negative values indicate a more likely binding model, and (ii) the confidence score, where a confidence score above 0.7 suggests a high probability of binding, a score between 0.5 and 0.7 indicates that binding is possible, and a score below 0.5 suggests a low likelihood of binding (Adasme et al, 2021). The protein-nucleic acid binding process is visualized in surface representation.

## CUT&RUN

CUT&RUN assays were performed using the CUT&RUN Assay Kit in accordance with the manufacturer's protocol. The DNA obtained from the assay was subjected to quantitative analysis. Each reaction was conducted in technical replicates, and experimental data were normalized using spike-in DNA. The primers used in this experiment were shown in Appendix Table S5.

## Establishment and treatment of 4-Nitroquinoline 1-oxide OLK animal model

To construct the OLK mouse model, 8-week-old male C57BL/6 mice (obtained from the Southern China Model Organisms Experimental Company) were continuously fed water containing 50 µg/mL 4-Nitroquinoline 1-oxide for 16 weeks (Yang et al, 2024). At week 16, the successful construction of the model was verified by visual assessment and HE staining of tongue tissue. Subsequently, 16 mice with typical leukoplakia lesions on the dorsum of the tongue were selected and randomly divided into four groups

($n = 4$) to receive different treatments. The control group (NC) was administered an intraperitoneal injection of co-solvents, while the Brusatol group (Brusatol) was given an intraperitoneal injection of Brusatol. The PDT group (PDT) underwent local PDT treatment after intraperitoneal injection of co-solvents. The combined treatment group (COMBO) was administered an intraperitoneal injection of Brusatol followed by PDT treatment. Brusatol was prepared as a clarified solution comprising 5% dimethyl sulfoxide (DMSO) and 95% corn oil. The experimental timeline was as follows: on days 1 to 3, mice in the Brusatol and COMBO groups were administered Brusatol (0.5 mg/kg) via intraperitoneal injection on a daily basis, while the remaining control mice were injected with an equal volume of co-solvent. On day 3, mice in the PDT and COMBO groups were treated with PDT as follows: following anesthesia, the tongue was injected locally with 20% 5-aminolevulinic acid (50 µL). After 1 h, the tongue lesion tissue was exposed to a red LED light at a wavelength of 635 nm for 2 min, with an output power of 50 mW and a cumulative energy fluence of 6 J/cm$^2$ (BHY-PDT-TT200S, Tianjin, China). Meanwhile, the NC and Brusatol groups were injected with saline at the same time point on the tongue. The mice were observed for a period of three weeks following the administration of the treatment, and were subsequently euthanised on the 24th day. Thereafter, the extent of the tongue lesions was evaluated, and the tissues were fixed for subsequent histological analysis. The efficacy of the treatment for OLK lesions was evaluated using the therapeutic effect score (Lin et al, 2022), while the degree epithelial dysplasia was assessed in accordance with the WHO Guidelines (2017). The epithelial dysplasia score quantifies the severity of dysplasia, assigning a score of 5 to invasive carcinoma, 4 to severe dysplasia/carcinoma in situ, 3 to moderate-to-severe dysplasia, 2 to moderate dysplasia, 1 to mild dysplasia, and 0 to the absence of dysplasia (Lin et al, 2022), while the pathological grade was assessed by two pathologists. To ensure the reliability of the results, the operators responsible for the intervention were aware of the grouping of mice, while the analysts responsible for the efficacy, pathological grading and statistical analysis were kept blind.

## Statistical analysis

Statistical comparisons were performed using unpaired two-sided Student's t-test, two-way ANOVA, or Mann–Whitney U test, depending on the data distribution, with a value of $p < 0.05$ indicating statistical significance. Correlation analyses were conducted using Pearson correlation coefficient and logistic regression analysis. The software used for statistical analyses included GraphPad Prism (version 9.1) and IBM SPSS (version 26). All experiments in this manuscript were repeated at least three times, $*p < 0.05$, $**p < 0.01$, $***p < 0.001$, $****p < 0.0001$. A list of antibodies and main reagents is provided in Appendix Table S6.

## Ethical approval and informed consent

This study was conducted in accordance with the principles outlined in the World Medical Association Declaration of Helsinki and the Belmont Report issued by the U.S. Department of Health and Human Services. Human tissue collection and experimental procedures were approved by the Ethics Committee of Sichuan University West China Hospital of Stomatology (WCHSIRB-D-

## The paper explained

### Problem

Oral leukoplakia (OLK) is a common precancerous lesion in the oral cavity and poses a risk of progressing to oral cancer. Photodynamic therapy (PDT) is a promising non-invasive treatment for OLK, but its effectiveness is limited by the development of resistance in many patients.

### Results

This study used single-cell RNA sequencing and tissue validation to reveal that PDT-resistant OLK tissues show elevated levels of phosphorylated NRF2 (p-NRF2). Activated NRF2 triggers antioxidant defense and keratinization programs, helping cells to neutralize PDT-induced oxidative stress. Mechanistically, NRF2 enhances WNT signaling by promoting the transcription of *CTNNB1*. Importantly, inhibition of NRF2 with Brusatol restored sensitivity to PDT in both cell and mouse models.

### Impact

The findings identify p-NRF2 as a potential biomarker for predicting PDT response in OLK patients. Combining NRF2 inhibition with PDT may overcome resistance and improve treatment outcomes, offering a new therapeutic strategy to reduce the risk of malignant transformation in patients with OLK.

2019-081) and the Ethics Committee of Peking University School and Hospital of Stomatology (PKUSSIRB-202162025). Written informed consent was obtained from all participants, including specific consent for the use of lesion photographs. Consent records have been securely retained.

All animal experiments were approved by the Ethical Review Board of West China Stomatological Hospital (WCHSIRB-D-2024-590) and were carried out in accordance with institutional guidelines for the care and use of laboratory animals.

## Data availability

The datasets produced in this study are available in the following databases: Raw sequence data: Genome Sequence Archive at the National Genomics Data Center, China National Center for Bioinformation, Beijing Institute of Genomics, Chinese Academy of Sciences, GSA-Human:HRA010961 (https://ngdc.cncb.ac.cn/gsa-human/browse/HRA010961).

The source data of this paper are collected in the following database record: biostudies:S-SCDT-10_1038-S44321-025-00256-w.

## Peer review information

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

## Acknowledgements

This study was supported by the National Natural Science Foundation of China (U19A2005, 82270986, 82470983, 82273320, 82330029 and 82472686), the CAMS Innovation Fund for Medical Sciences (CIFMS, 2019-I2M-5-004), and the 1·3·5 Project of Excellent Development of Discipline of West China Hospital of Sichuan University (No. ZYYC21001). The authors thank Qi Han (West China Hospital of Stomatology, Sichuan University) for his valuable guidance in pathology throughout the course of this study, as well as her assistance in the histopathological grading of the tissue samples. The authors also thank Ning Ji and Zhipeng Wang (State Key Laboratory of Oral Diseases, Sichuan University) for excellent assistance with microscopic imaging.

## Author contributions

**Tiantian Xu**: Conceptualization; Formal analysis; Validation; Investigation; Visualization; Methodology; Writing—original draft; Writing—review and editing. **Liang Zhong**: Conceptualization; Software; Formal analysis; Visualization; Methodology; Writing—review and editing. **Qianxi Liu**: Conceptualization; Formal analysis; Validation; Visualization; Methodology; Writing—original draft; Writing—review and editing. **Fei Wang**: Formal analysis; Validation; Visualization; Writing—original draft; Project administration; Writing—review and editing. **Wenjing Kuang**: Investigation; Methodology; Writing—review and editing. **Jiaqi Liang**: Investigation; Methodology; Writing—review and editing. **Dan Yang**: Investigation; Methodology; Writing—review and editing. **Xikun Zhou**: Conceptualization; Supervision; Funding acquisition; Methodology; Writing—review and editing. **Hongxia Dan**: Conceptualization; Formal analysis; Supervision; Funding acquisition; Investigation; Methodology. **Hang Zhao**: Conceptualization; Data curation; Supervision; Funding acquisition; Methodology. **Taiwen Li**: Data curation; Software; Supervision; Funding acquisition; Methodology; Writing—review and editing. **Xin Zeng**: Conceptualization; Resources; Data curation; Supervision; Funding acquisition; Methodology; Writing—review and editing. **Jing Li**: Conceptualization; Resources; Data curation; Supervision; Funding acquisition; Investigation; Methodology; Writing—review and editing. **Qianming Chen**: Conceptualization; Resources; Supervision; Funding acquisition; Methodology.

Source data underlying figure panels in this paper may have individual authorship assigned. Where available, figure panel/source data authorship is listed in the following database record: biostudies:S-SCDT-10_1038-S44321-025-00256-w.

## Disclosure and competing interests statement

The authors declare no competing interests.

# Expanded View Figures

**Figure EV1.   The UMAP visualization of cells from six patients with OLK.**

(**A**) Schematic diagram illustrating the inclusion and exclusion criteria for sample selection in scRNA-seq. (**B**) Distribution of mitochondrial gene expression percentage (percent_mt) in scRNA-seq. (**C**) Distribution of total RNA counts (nCount_RNA) in scRNA-seq. (**D**) Identification of highly variable genes based on scRNA-seq. Volcano plot showing highly variable genes, with each dot representing a single gene. Red dots indicate the 2,000 selected highly variable genes, among which the top 20 are labeled. Black dots represent genes not selected. (**E**) UMAP plot visualizing cells colored by clusters ($n = 14$ clusters). (**F**) UMAP plot visualizing cells colored by samples ($n = 6$ samples). (**G**) Bar plot displays the proportion of subclusters of epithelial cells in each group.

▶

                                                                          

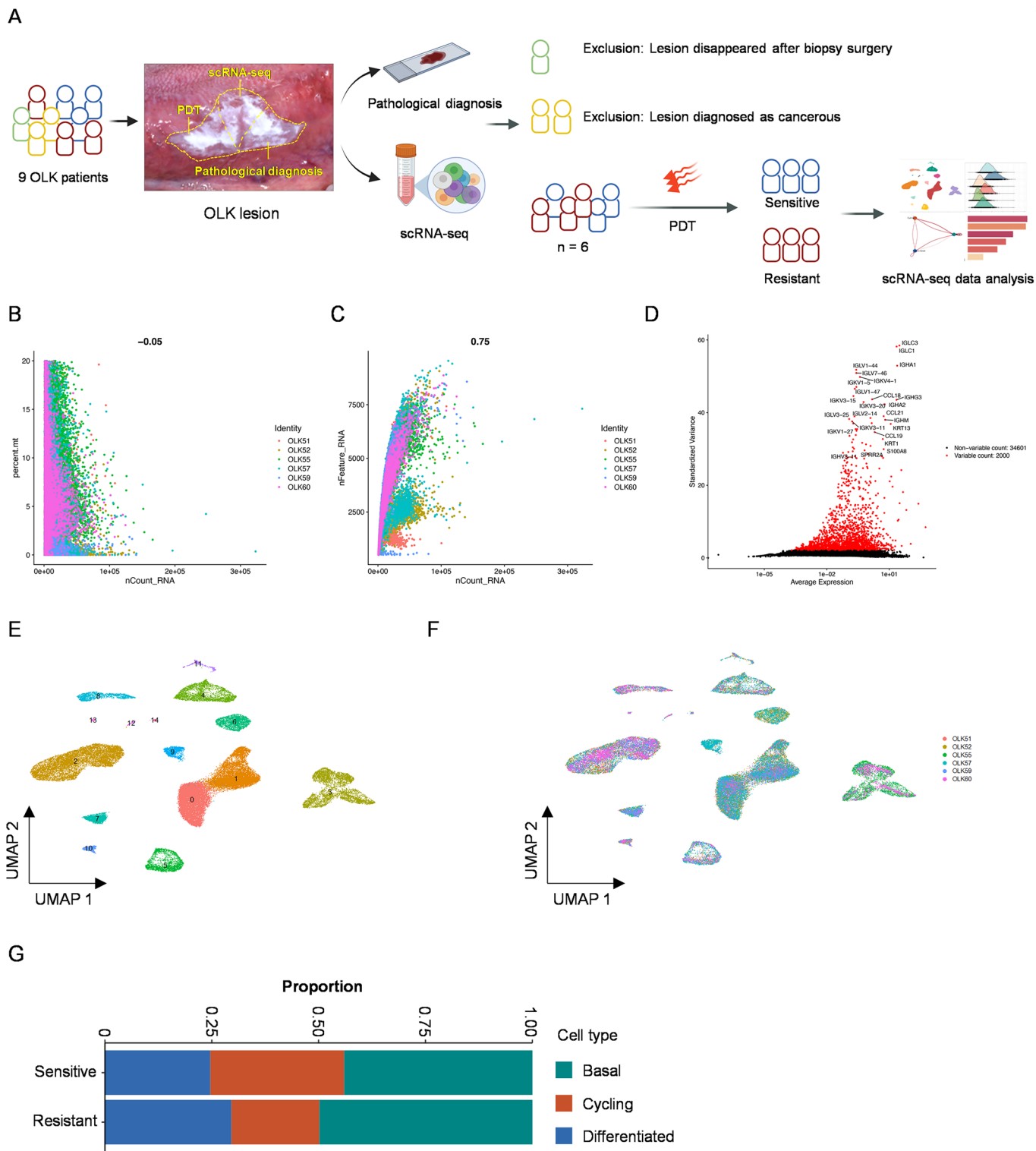

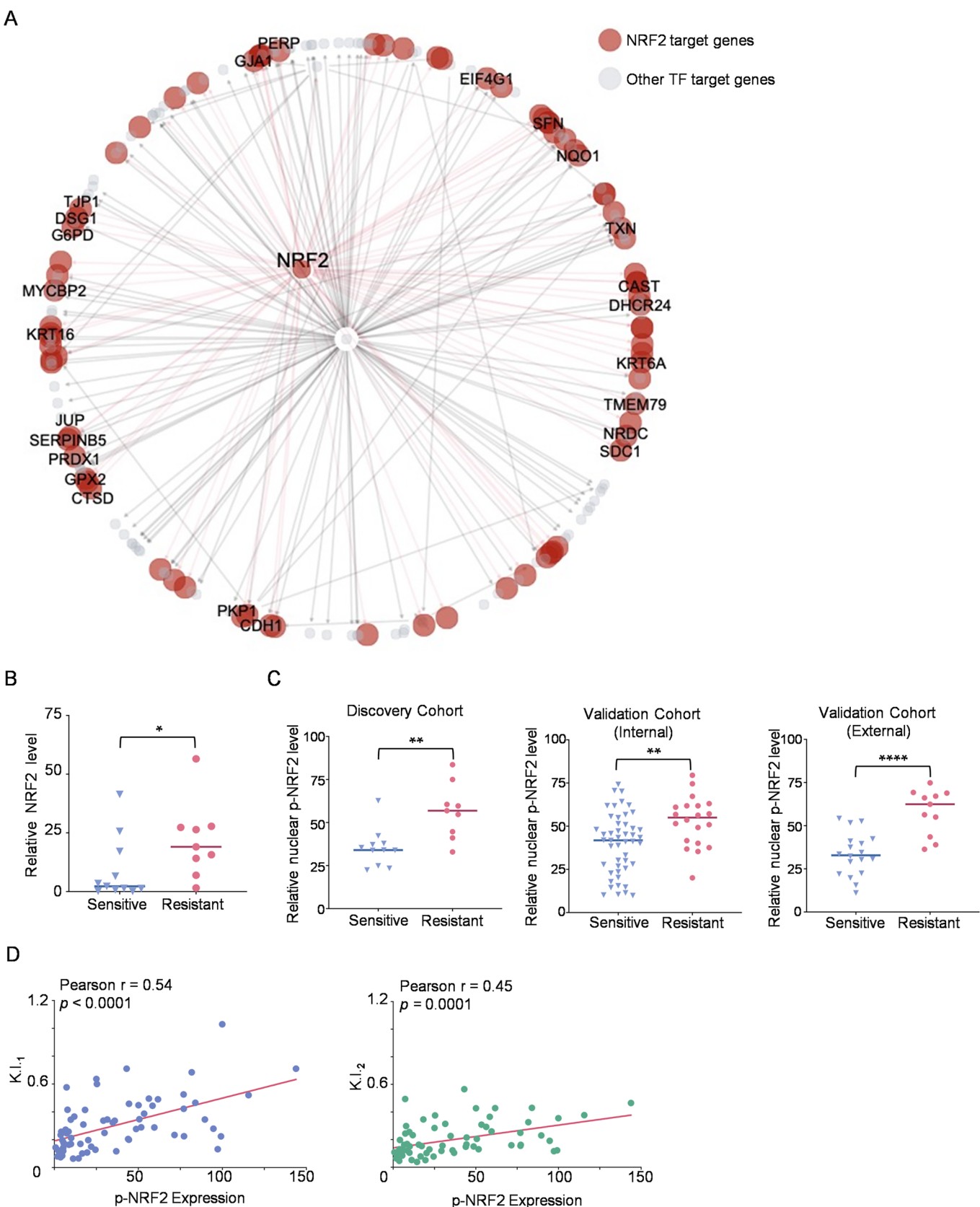

◀

**Figure EV2.   The expression of NRF2 and p-NRF2 in PDT-resistant OLK patients.**

(A) Network diagram of NRF2 downstream regulatory target genes. (B) Relative NRF2 level in PDT-sensitive ($n = 11$) and resistant ($n = 9$) OLK samples from the discovery cohort in IHC analysis (Student's t-test). (C) Relative nuclear p-NRF2 level in PDT-sensitive and resistant OLK samples from the three cohorts in IHC analysis. Sample sizes were: discovery cohort (sensitive: $n = 11$, resistant: $n = 9$), internal validation cohort (sensitive: $n = 48$, resistant: $n = 20$), and external validation cohort (sensitive: $n = 18$, resistant: $n = 11$). Statistical comparisons were performed using Student's t-test. (D) Correlation analysis of p-NRF2 expression and tissue K.I. in the internal validation cohort. Pearson correlation significance was assessed using Student's t-test ($n = 68$). Data information: In (B–D), each data point represents an individual patient (biological replicate). Significance is indicated as *$p < 0.05$, **$p < 0.01$, ****$p < 0.0001$. Exact $p$-values for these comparisons are provided in Appendix Table S1.

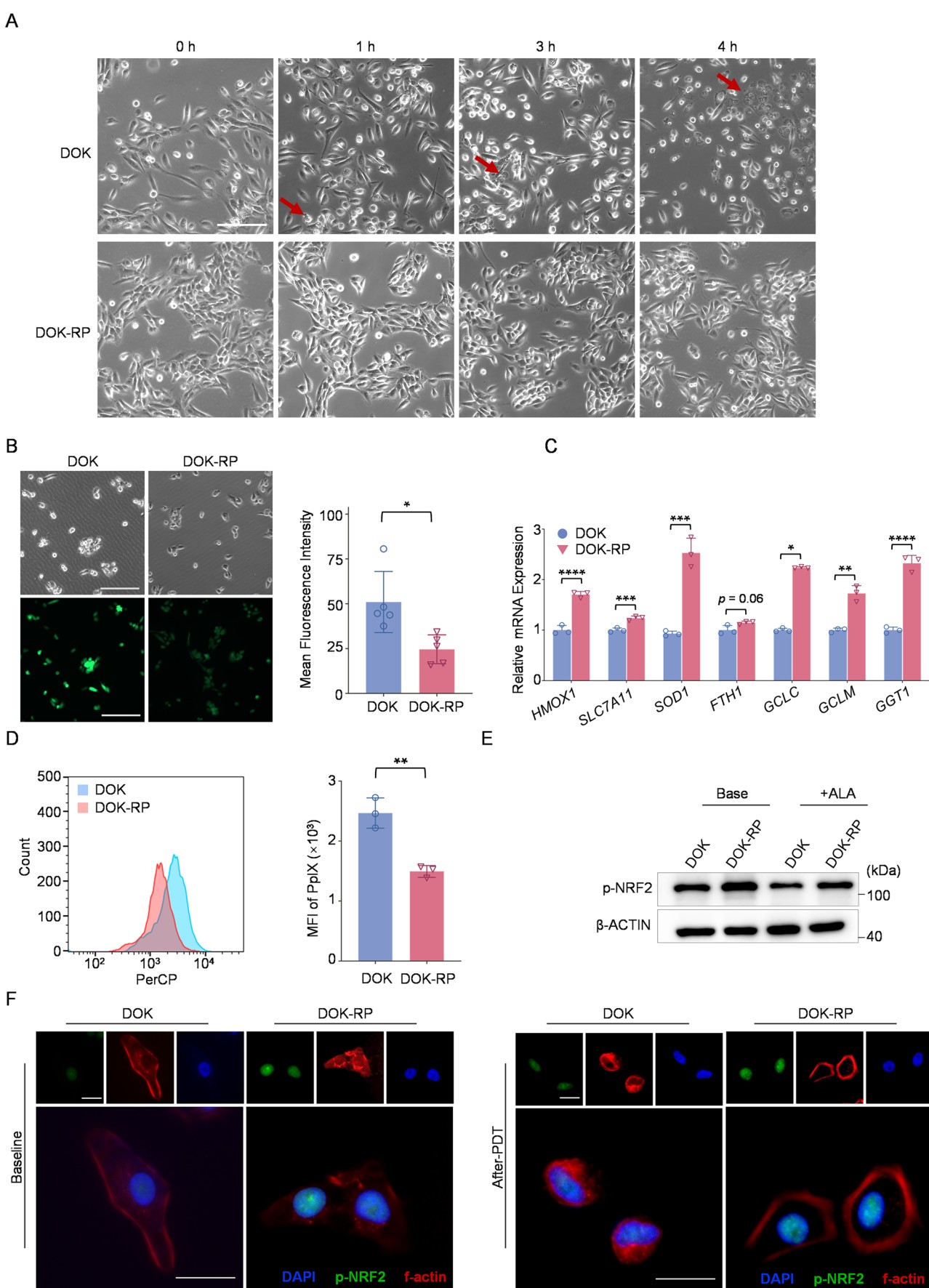

◀ **Figure EV3. Aberrant activation of NRF2 in PDT-resistant OLK cell model.**

(A) Representative images of morphological changes in DOK and DOK- RP cells after PDT across multiple time points. Scale bar: 50 μm. (B) The levels of ROS in PDT-treated DOK and DOK-RP cells were detected using CM-H$_2$DCFDA probes and observed under fluorescence microscopy. Data represent biological replicates ($n = 5$). Statistical results were presented in the right (Student's t-test). Scale bar: 100 μm. (C) Relative mRNA expression of antioxidant stress-related markers in DOK and DOK-RP cells. Data represent biological replicates ($n = 3$). Statistical comparison was performed using Student's t-test. (D) Comparison of PpIX uptake by DOK and DOK-RP cells detected by flow cytometry. The statistical results were presented in the right. Data represent biological replicates ($n = 3$). Statistical comparison was performed using Student's t-test. (E) Immunoblotting analysis of the expression of p-NRF2 in DOK and DOK-RP cells. (F) IF showing the expression of p-NRF2 in DOK and DOK-RP cells. Scale bar: 100 μm. Data information: In (B–D), data are presented as mean ± SD. Each data point represents an individual biological replicate. Significance is indicated as *$p < 0.05$, **$p < 0.01$, ***$p < 0.001$. Exact $p$-values for these comparisons are provided in Appendix Table S1.

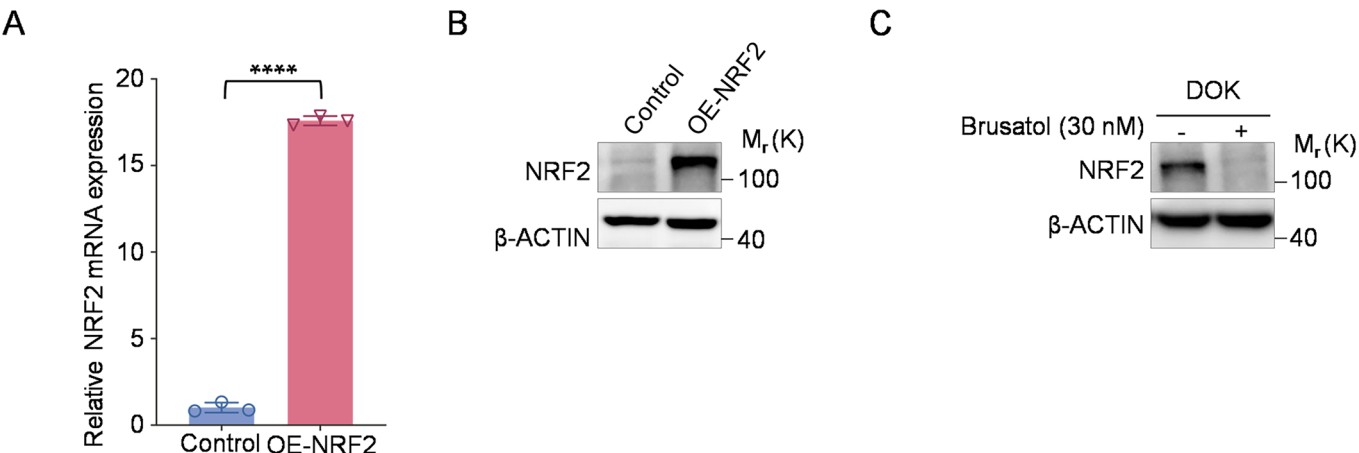

**Figure EV4. The expression of NRF2 in OLK cell models.**

(A) qPCR analyzes relative NRF2 mRNA expression in NRF2 overexpressing DOK cells. Data represent biological replicates ($n = 3$). Statistical comparison was performed using Student's t-test. (B) Immunoblotting analyzes the expression of NRF2 in NRF2-overexpressing DOK cell. (C) Immunoblotting analyzes the expression levels of NRF2 in DOK cells following Brusatol treatment. Data information: In (A), data are presented as mean ± SD. Each data point represents an individual biological replicate. Significance is indicated as ****$p < 0.0001$. Exact $p$-values for these comparisons are provided in Appendix Table S1.

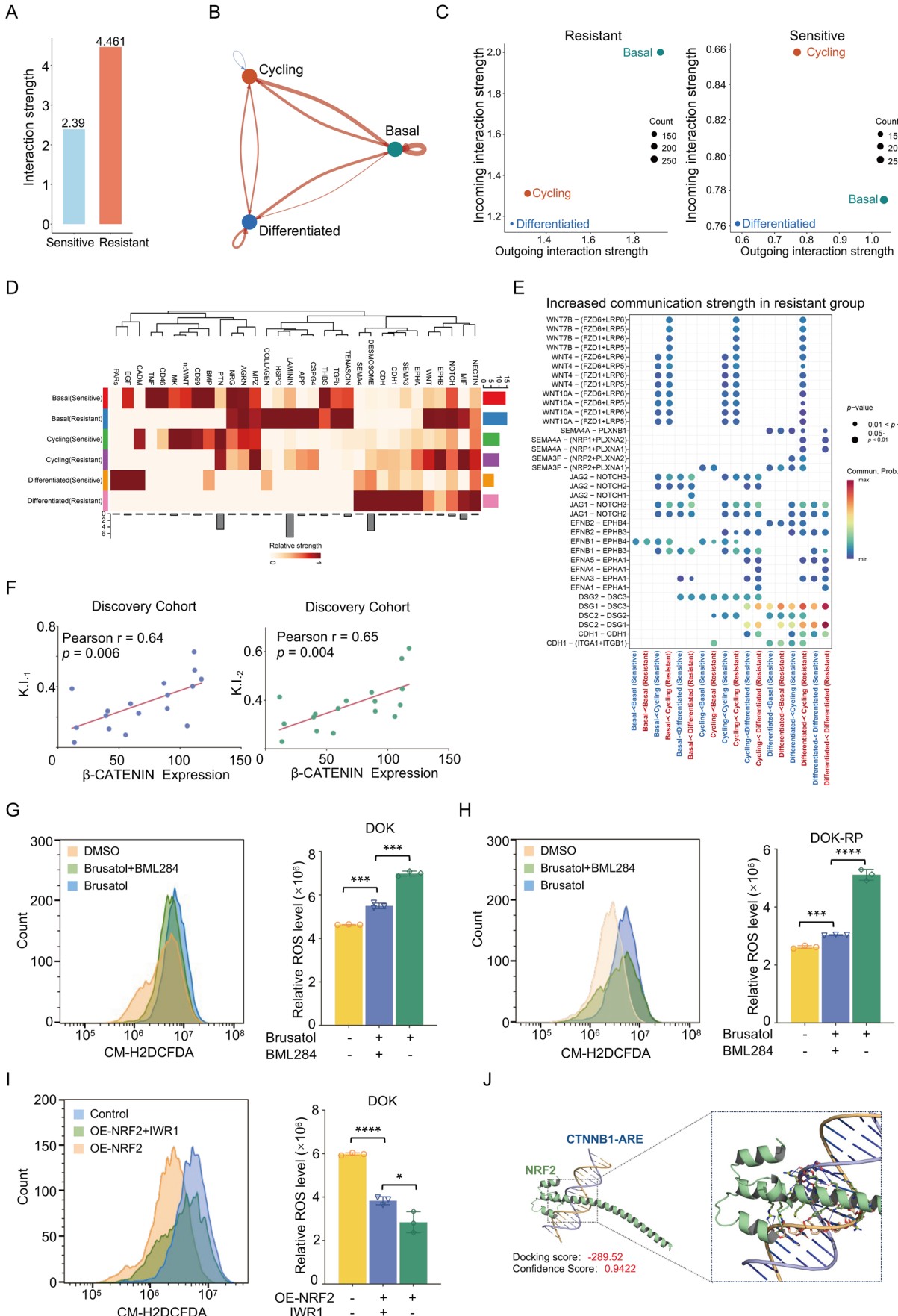

◀ **Figure EV5.    Contribution of WNT signaling pathway to PDT resistance in OLK cell model.**

(A) The bar plot showing cellular interaction strength in the sensitive and resistant groups. (B) The circle plot displaying the number of receptor/ligand pairs with significant differences between epithelial cell subtypes of the resistant group. (C) The bubble plot showing incoming/outgoing interaction strength among subclusters of epithelial cells in each group. (D) The heatmap displays intercellular communication signals of various epithelial cell subtypes in the sensitive and resistant groups. (E) The bubble plot displays the relative upregulation of ligand receptor signaling among epithelial cell subtypes in the OLK PDT-resistant group. Subpopulations include Basal (sensitive: $n = 1466$ cells; resistant: $n = 799$), Cycling (sensitive: $n = 1042$; resistant: $n = 330$), and Differentiated (sensitive: $n = 820$; resistant: $n = 474$). Statistical analysis was performed using a nonparametric permutation test. (F) Correlation between β-CATENIN and tissue K.I. Statistical significance for the Pearson correlation coefficient was assessed using Student's t-test ($n = 17$). (G) Flow cytometry showing the effect of combined treatment with BML284 and Brusatol on ROS levels in DOK cells after PDT treatment. Data represent biological replicates ($n = 3$). Statistical comparison was performed using Student's t-test. (H) Flow cytometry showing the effect of combined treatment with BML284 and Brusatol on ROS levels in DOK-RP cells after PDT treatment. Data represent biological replicates ($n = 3$). Statistical comparison was performed using Student's t-test. (I) Flow cytometry showing the effect of IWR1 on ROS levels in DOK cells overexpressing NRF2 after PDT treatment. Data represent biological replicates ($n = 3$). Statistical comparison was performed using Student's t-test. (J) Using HDOCK to predict the binding sites between NRF2 (green) and the ARE sequence (yellow and purple interactions) in the *CTNNB1* promoter. Data information: In (G–I), data are presented as mean ± SD. In (F–I), each data point represents an individual biological replicate. Significance is indicated as *$p < 0.05$, ***$p < 0.001$, ****$p < 0.0001$. Exact $p$-values for these comparisons are provided in Appendix Table S1.

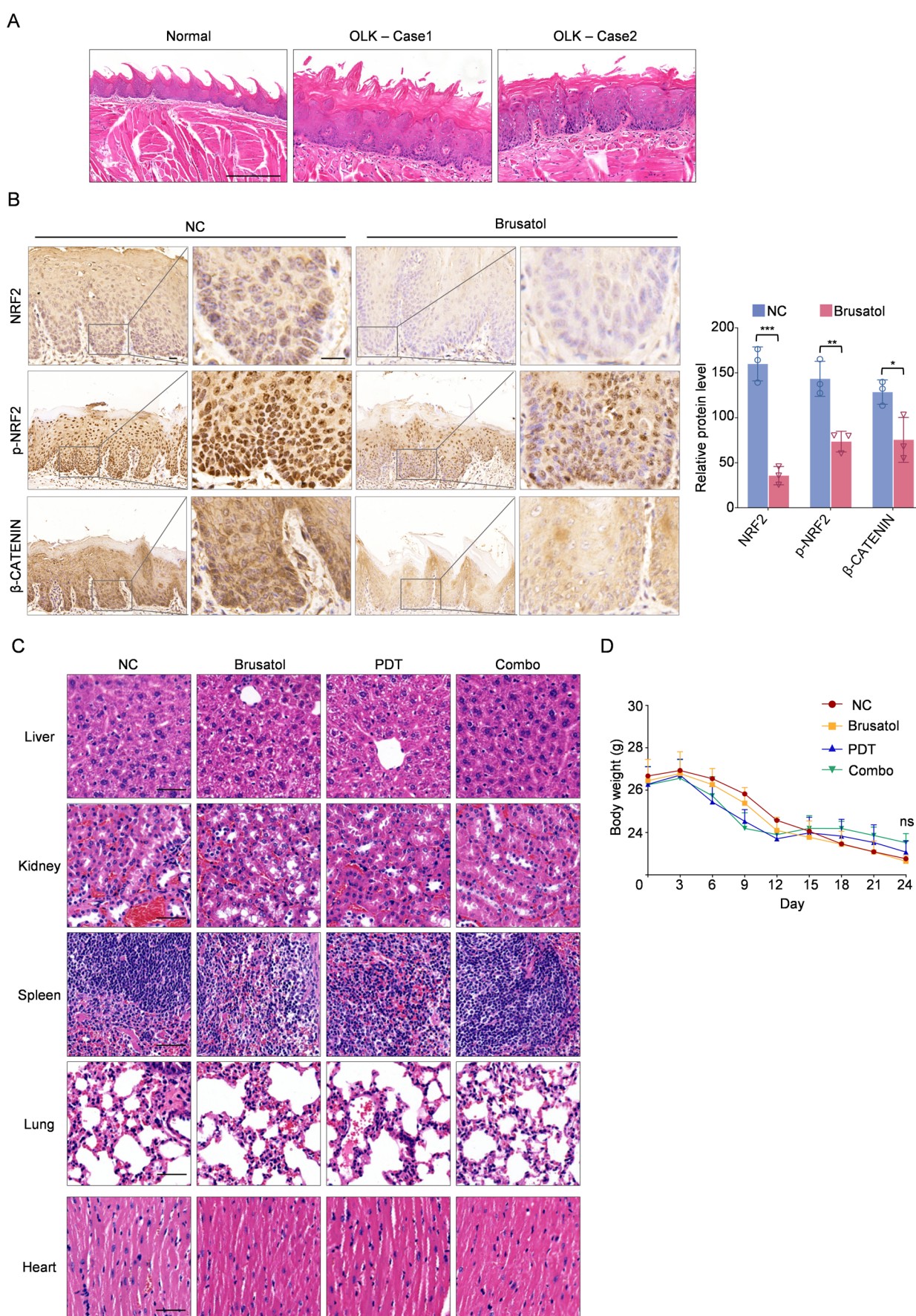

**Figure EV6.  Brusatol inhibited the expression of NRF2 and β-CATENIN in the OLK mouse model.**

(A) Representative HE images of tongue lesions from 4NQO-induced OLK mouse models at week 16 and normal tongue tissues from control mice. Scale bar: 200 μm. (B) Representative IHC images of NRF2, p-NRF2, and β-CATENIN expression in tongue tissues from Brusatol-treated and NC mice at day 3 (left, scale bar: 20 μm). Each group included $n = 3$ mice. Relative IHC expression levels of NRF2, p-NRF2, and β-CATENIN were quantified and compared between groups (right). Statistical comparison was performed using Student's t-test. (C) Representative HE images of the liver, kidney, spleen, lung, and heart from mice in each group. Scale bar: 50 μm. (D) Weight changes of mice in each group before and after treatment ($n = 4$ mice per group). Statistical comparison was performed using one-way ANOVA followed by Tukey's post hoc test. Data information: In (B, D), data are presented as mean ± SD. Each data point represents an individual biological replicate. Significance is indicated as *$p < 0.05$, **$p < 0.01$, ***$p < 0.001$, ns: not significant ($p \geq 0.05$). Exact p-values for these comparisons are provided in Appendix Table S1. Source data are available online for this figure.

