## [Peer Review File · EMBO Molecular Medicine]

NRF2 Modulates WNT Signaling Pathway to Enhance Photodynamic Therapy Resistance in Oral Leukoplakia

Tiantian Xu, Liang Zhong, Qianxi Liu, Fei Wang, Wenjing Kuang, Jiaqi Liang, Dan Yang, Xikun Zhou, Hongxia Dan, Hang Zhao, Taiwen Li, Xin Zeng, Jing Li, and Qianming Chen

Corresponding authors: Jing Li (lijing1984@scu.edu.cn) , Taiwen Li (litaiwen@scu.edu.cn), Xin Zeng (zengxin@scu.edu.cn)

Review Timeline:

Submission Date:	31st Jan 25
Editorial Decision:	6th Mar 25
Revision Received:	8th Apr 25
Editorial Decision:	2nd May 25
Revision Received:	9th May 25
Accepted:	16th May 25

Editor: Zeljko Durdevic

Transaction Report:

6th Mar 2025

Dear Dr. Li,

Thank you for the submission of your manuscript to EMBO Molecular Medicine. We have now received feedback from the three reviewers who agreed to evaluate your manuscript. All three referees recognize interest of the study but also raise important concerns that should be addressed in a major revision. If you would like to discuss further the points raised by the referees, I am available to do so via email or video. Let me know if you are interested in this option.

We would welcome the submission of a revised version within three months for further consideration. Please let us know if you require longer to complete the revision.

I look forward to receiving your revised manuscript.

Yours sincerely,

Zeljko Durdevic

Zeljko Durdevic
Senior Editor
EMBO Molecular Medicine

We require:

- 1) A .docx formatted version of the manuscript text (including legends for main figures, EV figures and tables). Please make sure that the changes are highlighted to be clearly visible.
- 2) Individual production quality figure files as .eps, .tif, .jpg (one file per figure). For guidance, download the 'Figure Guide PDF': (<https://www.embopress.org/page/journal/17574684/authorguide#figureformat>).
- 3) A .docx formatted letter INCLUDING the reviewers' reports and your detailed point-by-point responses to their comments. As part of the EMBO Press transparent editorial process, the point-by-point response is part of the Review Process File (RPF), which will be published alongside your paper.
- 4) A complete author checklist, which you can download from our author guidelines (<https://www.embopress.org/page/journal/17574684/authorguide#submissionofrevisions>). Please insert information in the checklist that is also reflected in the manuscript. The completed author checklist will also be part of the RPF.
- 5) Please note that all corresponding authors are required to supply an ORCID ID for their name upon submission of a revised manuscript.
- 6) It is mandatory to include a 'Data Availability' section after the Materials and Methods. Before submitting your revision, primary datasets produced in this study need to be deposited in an appropriate public database, and the accession numbers and

database listed under 'Data Availability'. Please remember to provide a reviewer password if the datasets are not yet public (see <https://www.embopress.org/page/journal/17574684/authorguide#dataavailability>).

12) Author contributions: You will be asked to provide CRediT (Contributor Role Taxonomy) terms in the submission system. These replace a narrative author contribution section in the manuscript.

13) A Conflict of Interest statement should be provided in the main text.

14) Every published paper now includes a 'Synopsis' to further enhance discoverability. Synopses are displayed on the journal webpage and are freely accessible to all readers. They include a short stand first (maximum of 300 characters, including space) as well as 2-5 one-sentences bullet points that summarizes the paper. Please write the bullet points to summarize the key NEW findings. They should be designed to be complementary to the abstract - i.e. not repeat the same text. We encourage inclusion of key acronyms and quantitative information (maximum of 30 words / bullet point). Please use the passive voice. Please attach

these in a separate file or send them by email, we will incorporate them accordingly.

15) Include a Reagents and Tools Table as part of the Methods section, which can be downloaded from our author guidelines (<https://www.embopress.org/page/journal/17574684/authorguide#structuredmethods>)

***** Reviewer's comments *****

Referee #1 (Comments on Novelty/Model System for Author):

In the manuscript entitled: " NRF2 Modulates WNT Signaling Pathway to Enhance Photodynamic Therapy Resistance in Oral Leukoplakia" the authors aimed to assess the role of single-cell RNA-sequencing (scRNA-seq) to identify cell types and markers that contribute to PDT resistance in OLK.

The authors found that NRF2 enhances PDT resistance by promoting CTNNB1 transcription, thereby modulating the WNT signaling pathway, reshaping the reactive oxygen species (ROS) response, and regulating cellular keratinization. Furthermore, in the 4NQO-induced OLK mouse model, the combination of NRF2 inhibitors with PDT effectively reversed OLK lesions and significantly normalized the mucosal tissue.

The authors concluded that phosphorylated NRF2 as a valuable biomarker for guiding PDT regimens in OLK patients, reveal NRF2's role in mediating PDT resistance via the WNT signaling pathway, and highlight NRF2 inhibition as a promising strategy to enhance PDT efficacy.

Referee #1 (Remarks for Author):

In general, the idea and innovation of this study regards the analysis of pathways and biomarkers in some oral diseases is interesting and novel because the role these aspects in medicine are validated but further studies on this topic could be an innovative issue in this field could be open a creative matter of debate in literature by adding new information. Moreover, there are few reports in the literature that studied this interesting topic with this kind of study design.

The study was well conducted by the authors; However, there are some concerns to revise that are described below.

The introduction section resumes the existing knowledge regarding the important factor linked with the impact mediators involved before and after oral diseases treatment.

However, as the importance of the topic, the reviewer strongly recommends, before a further re-evaluation of the manuscript, to update the literature through read, discuss and must cites in the references with great attention all of those recent interesting articles, that helps the authors to better introduce a similar oral disease form such as oral lichen planus and discuss the role of its local treatment that could improves several preneoplastic similar lesions: 1) doi: 10.1111/odi.13960 PMID: 34252252; 2) DOI: doi: 10.1111/odi.15156, PMID: 39402896

The authors should be better specified, at the end of the introduction section, the rationale of the study and the aim of the study. In the central section, should better clarify inclusions and exclusions criteria of the selected sample.

Please better state the results obtained in the abstract.

The discussion section appears well organized with the relevant paper that support the conclusions, even if the authors should better discuss the relationship regarding the by local inflammations that shares both patients with leucoplakia and liken in and risk of preneoplastic evolution that could improve the quality of life in oral diseases patients. The conclusion should reinforce in light of the discussions.

In conclusion, I am sure that the authors are fine clinicians who achieve very nice results with their adopted protocol. However, this study, in my view does not in its current form satisfy a very high scientific requirement for publication in this journal and requests a revision before a futher re-evaluation of the manuscript.

Minor Comments:

Abstract:

- Better formulate the abstract section by better describing the aim of the study

Introduction:

- Please refer to major comments

Discussion

- Please add a specific sentence that clarifies the results obtained in the first part of the discussion

Referee #2 (Remarks for Author):

With regard to " NRF2 Modulates WNT Signaling Pathway to Enhance Photodynamic Therapy Resistance in Oral Leukoplakia" a number of concepts are brought to the forefront of a difficult to treat condition. There are a number of points that could be addressed for this manuscript for publication purposes.

- 1) In the introduction there is a figure of malignant conversion rate of leukoplakia of 11.7 to over 30%. It is important to note that this data is often highly selected for the purpose of the publication. The generalized rates of conversion of leukoplakia throughout the literature are around 5% over 5 years, depending on how the leukoplakia is characterized (e.g. friction ridge leukoplakias which are part of the 3.5% incidence statistic, dysplastic leukoplakias, etc.) Would be good to review the literature and correct.
- 2) Correct terms like easy recurrence, easy scar formation. English or other qualifications of these statements should undergo careful review. For example, facial appearance is not affected by almost all of oral cancers, except large lip cancers. This requires revision.
- 3) Ki 67 indices for these cells are typically 1000 cells counted, cannot tell if this is the number of total cells counted by the measures in the methods.
- 4) For red light please explain, device, manufacturer, exposure time, etc. specified power settings not useful or rigorous enough for replicating data independently. Simply confirm the conditions for all experiments.
- 5) How did lentivirus transfection affect phenotype?
- 6) Hyperplasia and dysplasia are different histologies. Please confirm how each was qualified throughout the paper and how many pathologists evaluated each of the histologies. Are the oral mucosal specialists by board certified clinical pathologists or some other type of specialist or PhD subspecialty.
- 7) From an oral pathology perspective carcinoma in situ and severe dysplasia are often now the same thing by pathologic convention. Please use the scale and reference it for all of the hyperplasia and dysplasia criteria for the study. This may change the pathologic groupings though.
- 8) Irrespective of treatment type aboutn 30% of leukoplakia returns. In the discussion it talks about pdt being better. It may not be better, BUT it may be easier on patients to undergo repetitive procedures.
- 9) Throughout the results section and the discussion there is is reasonably high confident statements about NRF 2. Yet the human samples sequences are reasonably low in number. Please construct some sort of sample size calculations as well as a post hoc analysis of the sample numbers of both the human specimens and the mice to form some type of confidence intervals, false discovery rates etc. In general RNA seq one needs close to 30-35 samples to drive the FDR to a low enough probability to become validation for the next level of confidence in biomarkler studies.
- 10) Overall quite comprehensive analysis

Referee #3 (Comments on Novelty/Model System for Author):

To explore the molecular mechanisms underlying PDT resistance, a PDT-resistant OLK cell model was developed by following 15 rounds of PDT treatment. This is a new cellular model for the research of PDT resistance.

Referee #3 (Remarks for Author):

The manuscript by Xu et al. presents compelling and insightful data highlighting the critical role of Nrf2, especially phosphorylated Nrf2, in the development of resistance to photodynamic therapy (PDT) in oral leukoplakia (OLK). Their multicenter clinical findings demonstrate that p-Nrf2 is a robust predictor of PDT response among OLK patients. Additionally, they have established a disease-specific cellular model, providing a solid foundation for future research endeavors. Mechanistically, their study reveals that Nrf2 boosts β -catenin transcription, thereby activating the Wnt pathway and contributing to PDT resistance. Promising results from animal experiments further support combining PDT with Nrf2 inhibitors for treating OLK. In summary, this manuscript offers a rigorous and well-executed scientific investigation, introducing a novel perspective to the field, particularly given the limited existing research on this topic.

However, there are several minor points requiring clarification:

1. While the authors conducted quality control of single-cell data using parameters such as gene count, mitochondrial gene proportion, and RNA count, the article does not specify these parameters post-quality control for individual cells. Providing this

information is crucial for assessing the overall data quality.

2. In analyzing single-cell transcriptomic data, cell populations are typically annotated based on known marker expression. However, the identification of the keratinocyte subpopulation through functional annotation mentioned by the authors introduces confusion. Clarification regarding this annotation method would be beneficial.
3. During the Cut&Run experiment, DNA fragmentation into smaller pieces was performed. It remains unclear if the authors utilized agarose gel electrophoresis or other methods to verify effective DNA fragmentation.
4. The significance of the red arrows in Figure 3C is ambiguous. Clarifying whether these arrows denote mitochondrial structures within the EM image in either the figure legend or main text is necessary.
5. A scale bar is absent from the fluorescence image in Fig. S3B.
6. Regarding Figures 5I-K, which suggest NRF2 promotes transcription by binding to the ARE region of the catenin gene-an intriguing finding-further elucidation on the expression levels of the catenin gene in DOK and DOK-RP cells is needed to deepen understanding of the observed phenomena.

Referee #1 (Comments on Novelty/Model System for Author):

In the manuscript entitled: “NRF2 Modulates WNT Signaling Pathway to Enhance Photodynamic Therapy Resistance in Oral Leukoplakia” the authors aimed to assess the role of single-cell RNA-sequencing (scRNA-seq) to identify cell types and markers that contribute to PDT resistance in OLK.

The authors found that NRF2 enhances PDT resistance by promoting CTNNB1 transcription, thereby modulating the WNT signaling pathway, reshaping the reactive oxygen species (ROS) response, and regulating cellular keratinization. Furthermore, in the 4NQO-induced OLK mouse model, the combination of NRF2 inhibitors with PDT effectively reversed OLK lesions and significantly normalized the mucosal tissue. The authors concluded that phosphorylated NRF2 as a valuable biomarker for guiding PDT regimens in OLK patients, reveal NRF2's role in mediating PDT resistance via the WNT signaling pathway, and highlight NRF2 inhibition as a promising strategy to enhance PDT efficacy.

Authors' Response: We sincerely thank the reviewer for the comprehensive summary and positive recognition of our work. We are grateful for your appreciation of our findings and their potential significance in improving PDT outcomes for OLK patients.

Referee #1 (Remarks for Author):

In general, the idea and innovation of this study regards the analysis of pathways and biomarkers in some oral diseases is interesting and novel because the role these aspects in medicine are validated but further studies on this topic could be an innovative issue in this field could be open a creative matter of debate in literature by adding new information. Moreover, there are few reports in the literature that studied this interesting topic with this kind of study design. The study was well conducted by the authors; However, there are some concerns to revise that are described below.

Authors' Response: Thank you for your appreciation of our work and for your valuable comments. We appreciate your insights and are grateful for your positive feedback. We have carefully revised our manuscript based on your suggestions to further improve the quality of our study.

Comment #1. The introduction section resumes the existing knowledge regarding the important factor linked with the impact mediators involved before and after oral diseases treatment. However, as the importance of the topic, the reviewer strongly recommends, before a further re-evaluation of the manuscript, to update the literature through read, discuss and must cites in the references with great attention all of those recent interesting articles, that helps the authors to better introduce a similar oral disease form such as oral lichen planus and discuss the role of its local treatment that could improves several preneoplastic similar lesions: 1) doi: 10.1111/odi.13960 PMID: 34252252; 2) DOI: doi: 10.1111/odi.15156, PMID: 39402896

Authors' Response: We sincerely appreciate your insightful comments and valuable suggestions. Your feedback has significantly broadened the scope of our manuscript, enabling us to provide a more comprehensive discussion on the treatment and management of OPMDs. In response to your recommendation, we have carefully reviewed the suggested references and other recent literature. Accordingly, we have expanded the introduction section to include a discussion on the treatment strategies for other OPMDs as well as the potential application of PDT in these conditions. These additions enhance the contextual relevance of our study and further support the significance of investigating PDT resistance mechanisms in oral leukoplakia. Accordingly, we have revised the corresponding content in the **“Introduction” section, Paragraph 1, Lines 50–54, and Paragraph 2, Lines 67–74.** The revised text reads as follows:

(“Introduction” section, Paragraph 1, Lines 50–54) Notably, similar challenges persist in the management of other OPMDs. For instance, while local treatments such as corticosteroids and other pharmacological agents can effectively alleviate symptoms in oral lichen planus (OLP) [8, 9] and oral submucous fibrosis (OSF) [10], their efficacy in achieving long-term disease control and mitigating the risk of malignant transformation remains significantly limited [11, 12].

(“Introduction” section, Paragraph 2, Lines 67–74) Studies have shown that PDT not only holds significant promise for OLK treatment but also provides therapeutic

benefits in managing premalignant lesions in other OPMDs such as OLP [16]. Notably, PDT has been reported to effectively alleviate ulcerative symptoms in refractory erosive OLP [17] and improve mouth-opening capacity in OSF patients [18]. However, despite its potential in OPMDs management, the therapeutic efficacy of PDT remains suboptimal, with only 60% of OLK patients responding favorably [19, 20], while a substantial proportion exhibit resistance. This underscores the necessity for a deeper investigation into the underlying mechanisms of PDT resistance to improve treatment outcomes.

Comment #2. The authors should be better specified, at the end of the introduction section, the rationale of the study and the aim of the study.

Authors' Response: We sincerely appreciate your insightful comment. In response to your suggestion, we have revised the final part of the Introduction to explicitly articulate the rationale and objectives of our study. These revisions provide a clearer justification for conducting this research and enhance the logical coherence of the manuscript. We believe these modifications significantly improve the clarity and readability of the Introduction section. Thank you again for your valuable feedback.

Accordingly, we have revised the corresponding content in the **“Introduction” section, Paragraph 3, Line 81-86**. The revised text reads as follows:

Despite the increasing application of PDT in the management of OPMDs, its clinical translation remains constrained by an incomplete understanding of the molecular mechanisms underlying resistance and the lack of validated biomarkers for predicting treatment response and facilitating patient stratification. To address these limitations, this study systematically investigates the cellular and molecular determinants of PDT resistance in OLK through a comprehensive, multi-dimensional approach.

Comment #3. In the central section, should better clarify inclusions and exclusions criteria of the selected sample.

Authors' Response: We sincerely appreciate the reviewer's suggestion. In response, we have revised the manuscript to provide a more detailed and explicit description of the inclusion and exclusion criteria for the selected samples. These modifications enhance the clarity and transparency of our study design. We hope that this revision

addresses the reviewer's concern. Accordingly, we have revised the corresponding content in the **“Materials and methods” section, Paragraph “Clinical patients and ethical approval”, Line 99-109**. The revised text reads as follows:

Key inclusion criteria were: (i) age \geq 18 years and \leq 80 years; (ii) diagnosis of OLK according to the 2005 WHO criteria [11]; (iii) eligibility for PDT according to previously established indications [12, 13]. Key exclusion criteria were: (i) pregnancy or breastfeeding; (ii) presence of other oral mucosal diseases, current oral cancer, or a history of oral malignancy (iii) severe systemic illness or psychiatric disorders that could interfere with treatment adherence; (iv) inability to tolerate PDT due to medical contraindications or poor general condition. Additionally, the inclusion criteria for OLK lesion tissues in the scRNA-seq cohort also take lesion size into consideration. Specifically, the lesion must be sufficient to accommodate histopathological biopsy and single-cell sample preparation while retaining enough tissue for subsequent PDT treatment. All patients underwent the standard PDT protocol as described in prior studies [12, 13].

Comment #4. Please better state the results obtained in the abstract.

Authors' Response: We sincerely appreciate the reviewer's valuable suggestion. In the revised version, we have reorganized the abstract to more clearly and concisely present our key findings. These revisions enhance the readability of the manuscript and provide a more comprehensive summary of the study's results. Thank you again for your insightful feedback. We have revised the corresponding content in the **“Abstract” section, Line 24-38**. The revised text reads as follows:

Oral leukoplakia (OLK) is a common potentially malignant oral disorder with high risk of malignant transformation. While photodynamic therapy (PDT) offers a minimally invasive treatment for OLK, some patients showed resistance to PDT and the mechanisms remain unclear. This study aims to identify key regulatory pathways driving PDT resistance in OLK. Single-cell RNA sequencing on OLK samples (three PDT-sensitive, three PDT-resistant) revealed significant NRF2 upregulation in resistant tissues. Validation across two independent cohorts (n = 117) confirmed that p-NRF2 levels were significantly elevated in PDT-resistant cases, exhibiting strong

predictive power for treatment response (AUC > 0.8). Mechanistically, NRF2 promotes *CTNNB1* transcription, activates WNT signaling, modulates reactive oxygen species responses, and regulates keratinization, collectively contributing to PDT resistance. In a 4NQO-induced OLK mouse model, NRF2 inhibition combined with PDT effectively reversed OLK lesions and restored mucosal histology. These findings establish p-NRF2 as a valuable biomarker for guiding PDT regimens in OLK patients, reveal NRF2's role in mediating PDT resistance via the WNT signaling pathway, and highlight NRF2 inhibition as a promising strategy to enhance PDT efficacy.

Comment #5. The discussion section appears well organized with the relevant paper that support the conclusions, even if the authors should better discuss the relationship regarding the by local inflammations that shares both patients with leucoplakia and liken in and risk of preneoplastic evolution that could improve the quality of life in oral diseases patients.

Authors' Response: We sincerely appreciate your insightful feedback. In response, we have expanded our discussion on inflammation's role in OPMDs progression and the dual function of NRF2 in inflammation regulation. We also highlight its intricate interplay with PDT and the critical timing of NRF2 inhibition for optimal therapeutic efficacy. These revisions clarify the complex relationship between inflammation, NRF2, and disease progression, enhancing our understanding of targeted interventions to improve patient outcomes. Once again, we truly appreciate your valuable comments, which have significantly strengthened our study. Accordingly, we have revised the corresponding content in the **“Discussion” section, Paragraph 5, Line 593-597**. The revised text reads as follows:

Moreover, localized inflammation is widely reported in patients with OPMDs, such as OLK and OLP [57], playing a critical role in the progression of precancerous lesions [58]. Considering the critical role of NRF2 in inflammation regulation [59, 60], further studies are warranted to optimize its inhibition strategy and timing, with the aim of enhancing PDT responsiveness while minimizing the risk of malignant transformation.

Comment #6. The conclusion should reinforce in light of the discussions.

Authors' Response: We sincerely appreciate your valuable suggestion. In response, we have revised the conclusion section to reinforce our findings in light of the discussions. The revised conclusion now not only summarizes our key research results but also integrates important perspectives from the discussion, providing a more comprehensive synthesis of our study's significance. We believe these modifications enhance the clarity and impact of our conclusions. Accordingly, we have revised the corresponding content in the **"Discussion" section, Paragraph 6, Line 599-607.** The revised text reads as follows:

In conclusion, our investigation elucidates a key mechanism underlying PDT resistance in OLK, wherein elevated NRF2 expression activates the WNT signaling pathway, suppresses cytotoxic ROS production, and promotes epithelial keratinization, ultimately impairing PDT efficacy. Moreover, p-NRF2 expression was identified as a potential predictive biomarker for PDT responsiveness, providing a basis for individualized treatment strategies. These findings offer a new perspective on the role of NRF2 in mediating PDT resistance. Furthermore, this study proposes a promising therapeutic strategy combining NRF2 inhibition with PDT to enhance treatment efficacy, enabling the reversal of OLK lesions to normalized mucosa or low-grade, low-risk lesions and reducing the risk of malignant transformation into oral cancer.

Comment #7. In conclusion, I am sure that the authors are fine clinicians who achieve very nice results with their adopted protocol. However, this study, in my view does not in its current form satisfy a very high scientific requirement for publication in this journal and requests a revision before a further re-evaluation of the manuscript.

Authors' Response: We sincerely appreciate the reviewer's constructive comments and recognition of our clinical expertise and research findings. These revisions include strengthening the methodological details, deepening the discussion with relevant literature, and ensuring that our conclusions are well-supported by data. We hope that these improvements adequately address the reviewer's concerns and make our study more suitable for publication.

Comment #8. Minor Comments:

Abstract:

- Better formulate the abstract section by better describing the aim of the study

Authors' Response: Thank you for your suggestion. We have restructured and condensed the abstract to better clarify the study's aim and key findings. The revised abstract is as follows:

“Abstract” section, Line 24-38:

Oral leukoplakia (OLK) is the most common potentially malignant oral disorder and a precursor to oral squamous cell carcinoma. While photodynamic therapy (PDT) offers a minimally invasive treatment, its resistance mechanisms remain unclear. This study aims to identify key regulatory pathways driving PDT resistance in OLK. Single-cell RNA sequencing on six OLK samples (three PDT-sensitive, three PDT-resistant) revealed significant NRF2 upregulation in resistant tissues. Validation across two independent cohorts (n = 117) confirmed that phosphorylated NRF2 (p-NRF2) levels were significantly elevated in PDT-resistant cases, exhibiting strong predictive power for treatment response (AUC > 0.8). Mechanistically, NRF2 promotes *CTNNB1* transcription, activates WNT signaling, modulates reactive oxygen species responses, and regulates keratinization, collectively contributing to PDT resistance. In a 4NQO-induced OLK mouse model, NRF2 inhibition combined with PDT effectively reversed OLK lesions and restored mucosal histology. These findings establish p-NRF2 as a valuable biomarker for guiding PDT regimens in OLK patients, reveal NRF2's role in mediating PDT resistance via the WNT signaling pathway, and highlight NRF2 inhibition as a promising strategy to enhance PDT efficacy.

Comment #9. Introduction:

- Please refer to major comments

Authors' Response: Thank you for your comment. We have revised the Introduction section accordingly based on the major comments.

Comment #10

Discussion

- Please add a specific sentence that clarifies the results obtained in the first part of the discussion

Authors' Response: Thank you for your suggestion. We have revised the conclusion to include a specific sentence that clarifies the key results discussed in the first part of the discussion. Accordingly, we have revised the corresponding content in the **“Discussion” section, Paragraph 6, Line 599-607.**The revised text reads as follows:

In conclusion, our investigation elucidates a key mechanism underlying PDT resistance in OLK, wherein elevated NRF2 expression activates the WNT signaling pathway, suppresses cytotoxic ROS production, and promotes epithelial keratinization, ultimately impairing PDT efficacy. Moreover, p-NRF2 expression was identified as a potential predictive biomarker for PDT responsiveness, providing a basis for individualized treatment strategies. These findings offer a new perspective on the role of NRF2 in mediating PDT resistance. Furthermore, this study proposes a promising therapeutic strategy combining NRF2 inhibition with PDT to enhance treatment efficacy, enabling the reversal of OLK lesions to normalized mucosa or low-grade, low-risk lesions and reducing the risk of malignant transformation into oral cancer.

Referee #2 (Remarks for Author):

With regard to " NRF2 Modulates WNT Signaling Pathway to Enhance Photodynamic Therapy Resistance in Oral Leukoplakia" a number of concepts are brought to the forefront of a difficult to treat condition. There are a number of points that could be addressed for this manuscript for publication purposes.

Authors' Response: Thank you for your recognition and valuable suggestions. We have carefully revised and improved the manuscript accordingly, and we look forward to your further review.

Comment # 1. In the introduction there is a figure of malignant conversion rate of leukoplakia of 11.7 to over 30%. It is important to note that this data is often highly selected for the purpose of the publication. The generalized rates of conversion of leukoplakia throughout the literature are around 5% over 5 years, depending on how the leukoplakia is characterized (e.g. friction ridge leukoplakias which are part of the 3.5% incidence statistic, dysplastic leukoplakias, etc.) Would be good to review the literature and correct.

Authors' Response: Thank you for your valuable feedback. We appreciate your insightful comments and the opportunity to refine our manuscript. In response to your concern regarding the malignant conversion rate of leukoplakia presented in the introduction, we have extensively reviewed the relevant literature on this topic. Based on our comprehensive assessment, we have determined that the malignant transformation rate of oral leukoplakia ranges from 1% to 9.7%, depending on the classification criteria used in different studies [18-24]. Consequently, we have revised the corresponding statement in the introduction to reflect a more accurate and generalized estimate. We sincerely appreciate your suggestions, which have helped improve the clarity and accuracy of our manuscript.

Accordingly, we have revised the corresponding content in the **“Introduction” section, Paragraph1, Line 43-44.** The revised text reads as follows:

Its prevalence is reported to be 0.5%~3.46%, and its malignant transformation rate is 1% to 9.7% worldwide [20, 23, 24].

Comment # 2. Correct terms like easy recurrence, easy scar formation. English or other qualifications of these statements should undergo careful review. For example, facial appearance is not affected by almost all of oral cancers, except large lip cancers. This requires revision.

Authors' Response: Thank you for your insightful feedback. We appreciate your careful review and valuable suggestions. In response to your comment, we have revised our wording to ensure greater precision and avoid overstatement of the disease outcomes. Specifically, we have refined our descriptions of recurrence, scarring, and the impact on facial appearance to align more accurately with the clinical evidence. Accordingly, we have revised the corresponding content in the **“Introduction” section, Paragraph 1, Line 46-50**. The revised text reads as follows:

Traditional treatment methods include drug treatment, laser treatment, surgical resection, etc. [25]. **These treatment methods, whether applied individually or in combination, have several limitations, including limited efficacy, relatively high recurrence rates, resistance development, a risk of postoperative scarring, potential impairment of oral function, and, in some cases, negative effects on facial aesthetics and overall quality of life** [26].

Comment # 3. Ki 67 indices for these cells are typically 1000 cells counted, cannot tell if this is the number of total cells counted by the measures in the methods.

Authors' Response: Thank you for your valuable comment. We sincerely apologize for not clearly describing the method used to calculate the Ki67 index in the previous version of the manuscript. In our study, the Ki67 index was determined by randomly selecting five fields of view, each containing approximately 300–500 cells. The percentage of Ki67-positive cells was calculated for each field, and the final Ki67 index was obtained by averaging these values. As a result, the total number of cells analyzed ranged from approximately 1500 to 2500, which meets the standard requirement for Ki67 evaluation. To address this concern, we have revised the corresponding section in the **“Materials and methods” section, Paragraph “Immunohistochemical (IHC) staining”, line 157-160** to provide a clearer

description of our methodology. The revised text reads as follows:

The Ki67 index is determined by randomly selecting five fields of view, each containing approximately 300-500 cells. The percentage of Ki67-positive nuclei is calculated for each field, and the final Ki67 index is obtained by averaging these values [27].

Comment # 4. For red light please explain, device, manufacturer, exposure time, etc. specified power settings not useful or rigorous enough for replicating data independently. Simply confirm the conditions for all experiments.

Authors' Response: Thank you for your valuable comments. Regarding the red light device model and manufacturer used in this study, we have now provided additional details in the manuscript and supplementary materials. In our study, we use J/cm² (joules per square centimeter) as the metric to quantify the energy delivered during PDT treatment. J/cm² represents energy fluence or energy density, which describes the amount of energy received per unit area. This unit is widely used in PDT, laser treatment, and ultraviolet radiation exposure, serving as a critical parameter for determining light dosage, which directly affects treatment efficacy and cellular responses. To improve clarity, we have also included the calculation formula for energy fluence:

$$\text{Energy fluence (J/cm}^2\text{)} = \text{Power density (W/cm}^2\text{)} \times \text{Exposure time (s)}$$

In cell-based experiments, even when the same power settings and exposure time are used, the variation in the bottom surface area of culture plates (e.g., 6-well plates vs. 96-well plates) results in different spot sizes. Consequently, it is not feasible to standardize absolute irradiation time and power across different setups. However, since J/cm² is a universal metric for light dose measurement, it ensures comparability across different experimental conditions, which is why it is widely adopted in our study.

To enhance the reproducibility of our findings, we have supplemented the materials and methods section with detailed energy fluence density calculations and expanded descriptions of the PDT procedures used in both in vitro and in vivo

experiments, supported by relevant references. These revisions ensure that our data remain independently reproducible across different experimental setups and manufacturers' materials. Additionally, for clinical PDT procedures, we strictly adhere to the standardized protocol outlined in the OLK photodynamic therapy guidelines, and the detailed procedure can be found in the referenced guidelines. We appreciate the reviewer's suggestion, and we believe that these clarifications and revisions strengthen the transparency and reproducibility of our study. The revised text reads as follows:

“Materials and methods” section, Paragraph “Establishment of PDT-resistant OLK cell model” line 177-184: Cells undergoing ALA-PDT treatment were seeded in 96-well, 24-well, or 6-well plates and were pre-incubated in FBS-free medium containing 5-ALA (0.5 mM) in the dark for 4 hours. Following incubation, cells were exposed to red light at a wavelength of 635 nm(BHY-PDT-TT200S, Tianjin, China), with the laser spot size matched to the well diameter. The irradiation was performed according to the specified power settings [23], with energy fluence determined by the equation: Energy fluence (J/cm²) = Optical power density (W/cm²) × Exposure time (seconds). After irradiation, cells were further cultured until the designated time points for subsequent experiments.

“Materials and methods” section, Paragraph “Establishment and treatment of 4-Nitroquinoline 1-oxide OLK animal model” line 278-280: After 1 hour, the tongue lesion tissue was exposed to a red LED light at a wavelength of 635 nm for 2 minutes, with an output power of 50 mW and a cumulative energy fluence of 6 J/cm² (BHY-PDT-TT200S, Tianjin, China).

“Supplementary Tables” section, Table S6, line 51-53:

Experimental instrument		
PDT instrument for clinical application (Human)	LG (Chongqing, China)	LG-PDT-02
PDT instrument for experimental application (Cells and animals)	BHY (Tianjin, China)	BHY-PDT-TT200S

Comment # 5. How did lentivirus transfection affect phenotype?

Authors' Response:

Thank you for your valuable comment. We sincerely apologize for the lack of clarity in our previous description of this section. Lentivirus-mediated gene transduction is a commonly used technique in molecular biology and is widely applied in the construction of stable gene overexpression or knockdown cell models. Infection with lentivirus carrying an empty vector typically does not induce significant phenotypic changes in cells and is therefore frequently used as a control to evaluate the specific effects of the target gene. In this study, we used this technique to establish a DOK cell line with stable overexpression of NRF2. Due to its high transduction efficiency, stable integration into the host genome, and suitability for both dividing and non-dividing cells, lentiviral vectors have been widely utilized in gene editing and the development of stable cell lines in mammalian systems. Specifically, we employed a recombinant lentiviral vector (pLV-NRF2) carrying the NRF2 coding sequence to infect DOK cells, enabling stable genomic integration and sustained overexpression of NRF2. An empty vector (pLV) was used as a control to exclude any phenotypic effects resulting from viral infection itself. The infection was carried out at a multiplicity of infection (MOI) of 100, followed by selection with puromycin (5 µg/mL) for 15 days to eliminate uninfected cells and obtain a population stably expressing NRF2. The successfully selected cells were validated to overexpress NRF2 at both the mRNA and protein levels, as confirmed by RT-qPCR and Western blot analysis. To maintain the stability of the selected clones, low-dose puromycin (1 µg/mL) was continuously added to the culture medium to ensure selective growth of positive cells. In response to this concern, we have revised and supplemented the corresponding section in the Materials and Methods to clearly state the purpose and role of lentiviral transduction in our study. We greatly appreciate your insightful comment, which has helped improve the clarity and rigor of our work. The revised text reads as follows:

“Materials and methods” section, Paragraph “Construction of NRF2 stably overexpressing DOK cell line” line 190-193:

Recombinant lentiviruses (pLV-NRF2 and pLV) were obtained from VectorBuilder (Shanghai, China). DOK cells were infected with the recombinant lentiviral vector carrying the NRF2 coding sequence pLV-NRF2 to facilitate stable genomic integration and sustained overexpression of NRF2. An empty vector (pLV) was used as a negative control to exclude potential phenotypic effects associated with lentiviral infection. DOK cells were infected with these vectors (MOI = 100) and screened with puromycin (5 µg/mL) for 15 days. RT-qPCR and Immunoblotting (IB) analysis confirmed NRF2 overexpression at the mRNA and protein levels. Successfully transfected cell lines were maintained in complete medium with puromycin (1 µg/mL) for further analysis.

Comment # 6. Hyperplasia and dysplasia are different histologies. Please confirm how each was qualified throughout the paper and how many pathologists evaluated each of the histologies. Are the oral mucosal specialists by board certified clinical pathologists or some other type of specialist or PhD subspecialty.

Authors’ Response:

Thank you for your insightful comments. In the previous version of our manuscript, we did indeed misuse the term “hyperplasia.” We fully recognize that hyperplasia and dysplasia represent fundamentally different histological entities. Accordingly, we have carefully reviewed the entire manuscript and made the necessary corrections to ensure accurate terminology throughout. In our study, the histological evaluation was conducted by two board-certified clinical pathologists (Qi Han and Xin Zeng) who have extensive experience in diagnosing and assessing oral mucosal diseases. These pathologists have been working at West China Hospital of Stomatology for many years, specializing in the evaluation of oral mucosal lesions. We have acknowledged the valuable contributions of the two pathologists to this study in both the Acknowledgements and Author Contributions sections.

Comment # 7. From an oral pathology perspective carcinoma in situ and severe dysplasia are often now the same thing by pathologic convention. Please use the scale and reference it for all of the hyperplasia and dysplasia criteria for the study. This may change the pathologic groupings though.

Authors' Response:

Thank you for your thoughtful comment. In the previous version of our manuscript, we grouped samples with moderate-to-severe dysplasia under the category of severe dysplasia, following the approach adopted in several earlier studies that analyzed these two categories together. However, we acknowledge that this method may lack precision from a pathological perspective. To address this issue, we have re-evaluated the pathological grading of all samples in the revised manuscript according to the 2017 WHO classification of oral potentially malignant disorders. A more standardized classification system has now been applied, and the relevant sections of the manuscript have been updated accordingly to enhance the scientific rigor and consistency of our study. Our results have also been revised accordingly based on this classification criterion (**Response Figure 1**). We sincerely appreciate your valuable suggestion, which has helped us further improve our study design and methodology.

The revised text and figures are as follows:

“Materials and methods” section, Paragraph “Establishment and treatment of 4-Nitroquinoline 1-oxide OLK animal model” line 284-290:

The efficacy of the treatment for OLK lesions was evaluated using the therapeutics effect score [34], while the degree epithelial dysplasia was assessed in accordance with the WHO Guidelines (2017) [35]. The epithelial dysplasia score quantifies the severity of dysplasia, assigning a score of 5 to invasive carcinoma, 4 to severe dysplasia/carcinoma in situ, 3 to moderate-to-severe dysplasia, 2 to moderate dysplasia, 1 to mild dysplasia, and 0 to the absence of dysplasia [34], while the pathological grade was assessed by two pathologists.

“Result” section line 496-499:

Subsequently, we analyzed the efficacy scores and found that compared with the three control groups, the **COMBO** group significantly inhibited or even reversed the

progression of the lesions (Fig. 6D), and the degree of epithelial dysplasia was significantly lower (Fig. 6 E-G).

Response Figure 1(Figure 6 F-G in the manuscript)

Comment # 8. Irrespective of treatment type about 30% of leukoplakia returns. In the discussion it talks about pdt being better. It may not be better, BUT it may be easier on patients to undergo repetitive procedures.

Authors' Response: Thank you for your valuable suggestion. We have revised the discussion section to ensure a more objective presentation. Instead of implying that PDT is superior, we now emphasize its advantage in facilitating repeated treatments due to its minimally invasive nature and faster recovery, which may alleviate the physical and psychological burden on patients. We appreciate your insightful feedback in helping us improve the clarity and balance of our discussion. The revised text and figures are as follows:

“Discussion” section, Paragraph 1, line 512-516: Notably, regardless of the treatment modality, approximately 22% of OLK patients may experience recurrence [39]. In such cases, PDT offers a more favorable option than traditional surgical approaches due to its reduced trauma and faster recovery. These advantages not only mitigate the physical and psychological burden associated with repeated treatments but also improve the feasibility of long-term disease management [20].

Comment # 9. Throughout the results section and the discussion there is reasonably high confident statements about NRF2. Yet the human samples sequences are reasonably low in number. Please construct some sort of sample size calculations as

well as a post hoc analysis of the sample numbers of both the human specimens and the mice to form some type of confidence intervals, false discovery rates etc. In general RNA seq one needs close to 30-35 samples to drive the FDR to a low enough probability to become validation for the next level of confidence in biomarker studies.

Authors' Response:

We sincerely appreciate the reviewer's valuable comments. We fully acknowledge that, in conventional bulk RNA-seq studies, a dataset of approximately 30–35 samples is generally appropriate to effectively reduce the false discovery rate (FDR) and enhance the confidence level in biomarker validation. However, we would like to clarify that our study employs single-cell RNA sequencing (scRNA-seq), which differs significantly from bulk RNA-seq in terms of sample size requirements and experimental feasibility. scRNA-seq captures gene expression at the single-cell level, enabling identification of cellular heterogeneity, with each cell treated as a single sample and a sufficient number of biological replicates.

While scRNA-seq typically involves fewer biological samples compared to bulk RNA-seq, this can have implications for statistical power. Although a larger sample size would improve the robustness of our findings, it also introduces higher technical and financial burdens. Consequently, some studies, including ours, employ smaller sample sizes in the exploratory stage [31-33]. To ensure the reliability of our conclusions, we carefully validated all key findings using multiple independent experimental approaches.

For the clinical cohort, we included three independent research cohorts from multiple centers, comprising a total of 117 patients who received standardized PDT treatment. The cohorts included an internal exploration cohort (20 patients), an internal validation cohort (68 patients), and an external validation cohort (29 patients). The clinical and pathological data were meticulously recorded and analyzed, strengthening the rigor and clinical relevance of our study.

The sample size estimation for scRNA-seq data differs from traditional RNA-seq and other high-throughput data due to the nature of single-cell sequencing, where each cell represents independent data with typically higher variability. Therefore,

sample size estimation for scRNA-seq requires careful consideration of several factors, such as the number of cells, within-group variability, and experimental design. This introduces significant challenges in estimating the sample size for scRNA-seq studies. In terms of other sample size estimation, we performed calculations and power analyses for key data, which are summarized below:

We used G*Power software to estimate the sample size. This estimation is based on the effect size d , significance level α , statistical power $(1-\beta)$, and the type of test. The corresponding formula is as follows.

Effect Size (Cohen's d):

$$d = \frac{|\mu_1 - \mu_2|}{s}$$

μ_1, μ_2 : Group means

s : Pooled standard deviation:

$$s = \sqrt{\frac{|s_1^2 + s_2^2|}{2}}$$

Sample Size Calculation Formula :

$$n = \frac{2(Z_{1-\alpha} + Z_{1-\beta})^2}{d^2}$$

$Z_{1-\alpha}$: Z-value for significance level

$Z_{1-\beta}$: Z-value corresponding to statistical power

d : Effect size (Cohen's d)

In the actual calculation, we set the significance level α to 0.05 and the statistical power $(1-\beta)$ to 0.8 or 0.9. Based on the above formula, we either estimate the required sample size or calculate the statistical power using the actual sample size.

Figure 1G, I- Relative p-NRF2 level in internal/external validation cohort

	internal validation cohort	external validation cohort
Sensitive	25.73±29.02 (95% CI: 17.30, 34.15)	61.50±29.17 (95% CI: 47.01, 76.02)
Resistant	60.65±31.70	110.3±28.90

	(95% CI: 45.81, 75.48).	(95% CI: 90.87, 129.7)
α	0.05	0.05
Power(1- β)	0.9	0.9
Effect size	1.148	1.680
Sample size estimation (Sensitive vs. Resistant)	21 vs. 11	11 vs. 5
Actual sample size (Sensitive vs. Resistant)	48 vs. 20	18 vs. 11
Actual power	0.9146	0.9057
Power analysis diagram		
As the analysis shows, the sample sizes in both cohorts meet the requirements, with power values greater than 0.8, indicating good reliability.

Figure 2 H, J ROC curve analysis of p-NRF2 for predicting PDT efficacy

In ROC curve analysis, the AUC (Area Under the Curve) value reflects the model's ability to distinguish between positive and negative samples. AUC values closer to 1 indicate better performance, while AUC = 0.5 suggests no discriminative ability (similar to random guessing). AUC > 0.8 generally indicates strong discriminative ability, and AUC < 0.5 suggests the model performs worse than random guessing. Thus, the AUC serves as both a performance metric and an indicator of the model's effectiveness in distinguishing between categories. As shown in Figure 2 H and J, the AUC values for the internal and external validation cohorts were 0.825 (95% CI: 0.727, 0.923) and 0.884 (95% CI: 0.684, 0.911), respectively, indicating strong performance. The AUC values reflect the model's ability to distinguish between positive and negative samples, with higher values indicating better model performance.

For the animal experiments, we used the epithelial dysplasia score as the primary outcome measure for sample size estimation.

	NC	Brus	PDT	COMBO
Mean±SD	4.5±0.5774	4.375±0.9465	3.75±0.95	2.25±0.5
95% CI	(3.581, 5.419)	(2.869, 5.881)	(2.227, 5.273)	(1.454, 3.046)

Power analysis:

	NC vs. COMBO	Brus vs. COMBO	PDT vs. COMBO
α	0.05	0.05	0.05
Power(1- β)	0.8	0.8	0.8
Effect size	4.1659	2.8074	2.009
Sample size estimation	2	3	4
Actual sample size	4	4	4
Actual power	0.8269	0.8759	0.8046
Power analysis diagram			

As the analysis shows, the sample sizes in all comparisons meet the requirements, with power values greater than 0.8, indicating good reliability.

Although we have made every effort to enhance the scientific robustness of our study, we acknowledge that there is still room to optimize the sample size. In future research, if conditions permit, we will strive to expand the sample size to further improve statistical power and the robustness of our conclusions. We sincerely appreciate the reviewer's comments and hope that our explanation can address any concerns.

Comment # 10. Overall quite comprehensive analysis

Authors' Response: Thank you for your positive evaluation. We sincerely appreciate your recognition of our comprehensive analysis. We have carefully addressed all reviewer comments and made the necessary revisions to further enhance the clarity and rigor of our study. We hope that our revisions meet your expectations and look forward to your valuable feedback.

Referee #3 (Comments on Novelty/Model System for Author):

To explore the molecular mechanisms underlying PDT resistance, a PDT-resistant OLK cell model was developed by following 15 rounds of PDT treatment. This is a new cellular model for the research of PDT resistance.

Referee #3 (Remarks for Author):

The manuscript by Xu et al. presents compelling and insightful data highlighting the critical role of Nrf2, especially phosphorylated Nrf2, in the development of resistance to photodynamic therapy (PDT) in oral leukoplakia (OLK). Their multicenter clinical findings demonstrate that p-Nrf2 is a robust predictor of PDT response among OLK patients. Additionally, they have established a disease-specific cellular model, providing a solid foundation for future research endeavors. Mechanistically, their study reveals that Nrf2 boosts β -catenin transcription, thereby activating the Wnt pathway and contributing to PDT resistance. Promising results from animal experiments further support combining PDT with Nrf2 inhibitors for treating OLK. In summary, this manuscript offers a rigorous and well-executed scientific investigation, introducing a novel perspective to the field, particularly given the limited existing research on this topic.

Authors' Response: Thank you very much for your positive and encouraging feedback. We are truly grateful for your recognition of our work and your thoughtful summary of our findings. Your comments affirm the significance of our study and motivate us to further explore the role of p-Nrf2 in OLK and its potential as a therapeutic target. We sincerely appreciate your support and are committed to continuing rigorous and meaningful research in this area.

Comment # 1. However, there are several minor points requiring clarification: While the authors conducted quality control of single-cell data using parameters such as gene count, mitochondrial gene proportion, and RNA count, the article does not specify these parameters post-quality control for individual cells. Providing this information is crucial for assessing the overall data quality.

Authors' Response:

Thank you for your insightful comment. In response to your suggestion, we have now provided detailed descriptions of the quality control assessment of the scRNA-seq data. Specifically, we evaluated key cellular features, including the number of detected genes (*nFeature_RNA*), total RNA counts (*nCount_RNA*), and the proportion of mitochondrial genes (*percent_mt*). The results showed that *nFeature_RNA* was predominantly concentrated between 200 and 8000 across most cells, while *nCount_RNA* exhibited a normal distribution without abnormally high expression. Additionally, *percent_mt* was below 20% in the majority of cells, indicating high data quality with minimal influence from low-quality or apoptotic cells (**Response Figure 2**). These parameters confirm that the dataset meets the necessary criteria for subsequent analyses. We appreciate your valuable feedback, which has helped us enhance the clarity and transparency of our data quality assessment. The revised text and figures are as follows:

“Results” section, Paragraph 1, line 326-333: We conducted rigorous quality control of the scRNA-seq data by assessing key cellular feature metrics. The number of detected genes per cell (*nFeature_RNA*) predominantly ranged between 500 and 8000, and the total RNA counts (*nCount_RNA*) displayed a normal distribution without evidence of abnormally high expression levels. Additionally, the proportion of mitochondrial gene content (*percent_mt*) was below 20% in the vast majority of cells, indicating minimal contamination by low-quality or apoptotic cells. Collectively, these parameters demonstrate that the dataset meets the quality thresholds necessary for reliable downstream analyses (**Supplementary Fig. 1B-D**).

Response Figure 2 (Figure S1 B-D in the manuscript)

Comment # 2. In analyzing single-cell transcriptomic data, cell populations are typically annotated based on known marker expression. However, the identification of the keratinocyte subpopulation through functional annotation mentioned by the authors introduces confusion. Clarification regarding this annotation method would be beneficial.

Authors' Response: We sincerely apologize for the misinterpretation in our annotation of epithelial cells. In fact, we classified epithelial cells into basal, cycling, and differentiated subpopulations based on the expression of known marker genes of oral stratified squamous epithelium, rather than through functional analysis. These marker genes are associated with the function of keratinized epithelial cells, To clarify this, we have revised the original text as follows:

“Results” section, Paragraph 1, line 340-343: Next, we further re-clustered the epithelial cells and categorized them into basal, cycling, and differentiated subpopulations based on marker genes characteristic of oral stratified squamous epithelial cells. These marker genes correspond to the well-established functional roles of cells within the epidermis (Fig. 1G and Supplementary Fig. 1G).

Comment # 3. During the Cut&Run experiment, DNA fragmentation into smaller pieces was performed. It remains unclear if the authors utilized agarose gel electrophoresis or other methods to verify effective DNA fragmentation.

Authors' Response:

Thank you for your valuable suggestion. In fact, in the previous experiment, we routinely used agarose gel electrophoresis to verify the effectiveness of DNA fragmentation in the Cut&Run experiment. The figure shows that the DNA was successfully fragmented into 100–200 bp fragments (**Response Figure 3**), which meets the requirements of the Cut&Run experiment. We appreciate your insightful feedback.

Response Figure 3

Comment # 4. The significance of the red arrows in Figure 3C is ambiguous. Clarifying whether these arrows denote mitochondrial structures within the EM image in either the figure legend or main text is necessary.

Authors' Response: Thank you for your suggestion. We have clarified the significance of the red arrows by specifying in the figure legend that they indicate mitochondrial structures in the EM image. We appreciate your careful review and valuable feedback. The revised text and figures are as follows:

“Figure legend” section, Figure 3C : The red arrows indicate mitochondrial structures in the transmission electron microscopy image (scale bar, 2 μm).

Comment # 5. A scale bar is absent from the fluorescence image in Fig. S3B.

Authors' Response: Thank you for your careful review. We have added the missing scale bar to the fluorescence image in Fig. S3B. We appreciate your attention to detail and your valuable feedback.

Comment # 6. Regarding Figures 5I-K, which suggest NRF2 promotes transcription by binding to the ARE region of the catenin gene-an intriguing finding-further elucidation on the expression levels of the catenin gene in DOK and DOK-RP cells is needed to deepen understanding of the observed phenomena.

Authors' Response:

Thank you for your insightful suggestion. In response to your comment, we have supplemented the expression level data of the catenin gene in DOK and DOK-RP cells to further elucidate the observed phenomena. As shown in the figure, compared to DOK cells, the mRNA expression level of CTNNB1 is upregulated in DOK-RP cells(**Response Figure 4**). We appreciate your valuable feedback, which has helped us strengthen our study. The revised text and figures are as follows:

“Results” section, , line 485-486: Meanwhile, *CTNNB1* mRNA expression is upregulated in DOK-RP cells compared to DOK cells (**Fig. 5L**)

Response Figure 4 (Figure 5L in the manuscript)

Response reference:

1. Polizzi, A., et al., *Analysis of the response to two pharmacological protocols in patients with oral lichen planus: A randomized clinical trial*. *Oral Dis*, 2023. **29**(2): p. 755-763.
2. Polizzi, A., et al., *Impact of Topical Fluocinonide on Oral Lichen Planus Evolution: Randomized Controlled Clinical Trial*. *Oral Dis*, 2024.
3. Tang, J., et al., *Oral submucous fibrosis: pathogenesis and therapeutic approaches*. *Int J Oral Sci*, 2025. **17**(1): p. 8.
4. Lodi, G., et al., *Interventions for treating oral lichen planus: a systematic review*. *Br J Dermatol*, 2012. **166**(5): p. 938-47.
5. Sun, S.L., et al., *Topical calcineurin inhibitors in the treatment of oral lichen planus: a systematic review and meta-analysis*. *Br J Dermatol*, 2019. **181**(6): p. 1166-1176.
6. Binnal, A., et al., *Photodynamic therapy for oral potentially malignant disorders: A systematic review and meta-analysis*. *Photodiagnosis Photodyn Ther*, 2022. **37**: p. 102713.
7. Rakesh, N., et al., *Clinical evaluation of photodynamic therapy for the treatment of refractory oral Lichen planus – A case series*. *Photodiagnosis and Photodynamic Therapy*, 2018. **24**: p. 280-285.
8. Wan, W., et al., *5-aminolevulinic acid photodynamic therapy for extensive oral leukoplakia with concomitant oral submucous fibrosis: A case report*. *Photodiagnosis Photodyn Ther*, 2023. **41**: p. 103203.
9. Li, Y., et al., *Photodynamic therapy in the treatment of oral leukoplakia: A systematic review*. *Photodiagnosis Photodyn Ther*, 2019. **25**: p. 17-22.
10. Jing, Y., et al., *Clinical efficacy of photodynamic therapy of oral potentially malignant disorder*. *Photodiagnosis Photodyn Ther*, 2024. **46**: p. 104026.
11. Warnakulasuriya, S., N.W. Johnson, and I. van der Waal, *Nomenclature and classification of potentially malignant disorders of the oral mucosa*. *J Oral Pathol Med*, 2007. **36**(10): p. 575-80.
12. Chen, Q., et al., *Photodynamic therapy guidelines for the management of oral leukoplakia*. *Int J Oral Sci*, 2019. **11**(2): p. 14.
13. Yang, D., et al., *Ferroptosis Induction Enhances Photodynamic Therapy*

- Efficacy for OLK*. J Dent Res, 2024. **103**(12): p. 1227-1237.
14. Louisy, A., E. Humbert, and M. Samimi, *Oral Lichen Planus: An Update on Diagnosis and Management*. Am J Clin Dermatol, 2024. **25**(1): p. 35-53.
 15. Sun, Y., et al., *Immunosuppression Induced by Chronic Inflammation and the Progression to Oral Squamous Cell Carcinoma*. Mediators Inflamm, 2016. **2016**: p. 5715719.
 16. Cuadrado, A., et al., *Transcription Factor NRF2 as a Therapeutic Target for Chronic Diseases: A Systems Medicine Approach*. Pharmacol Rev, 2018. **70**(2): p. 348-383.
 17. Lu, M.C., et al., *The Keap1-Nrf2-ARE Pathway As a Potential Preventive and Therapeutic Target: An Update*. Med Res Rev, 2016. **36**(5): p. 924-63.
 18. Guan, J.Y., et al., *Malignant transformation rate of oral leukoplakia in the past 20 years: A systematic review and meta-analysis*. J Oral Pathol Med, 2023. **52**(8): p. 691-700.
 19. Pinto, A.C., et al., *Malignant transformation rate of oral leukoplakia-systematic review*. Oral Surg Oral Med Oral Pathol Oral Radiol, 2020. **129**(6): p. 600-611.e2.
 20. Schepman, K.P., et al., *Malignant transformation of oral leukoplakia: a follow-up study of a hospital-based population of 166 patients with oral leukoplakia from The Netherlands*. Oral Oncol, 1998. **34**(4): p. 270-5.
 21. Shearston, K., et al., *Malignant transformation rate of oral leukoplakia in an Australian population*. J Oral Pathol Med, 2019. **48**(7): p. 530-537.
 22. Wang, T.Y., et al., *Malignant transformation of Taiwanese patients with oral leukoplakia: A nationwide population-based retrospective cohort study*. J Formos Med Assoc, 2018. **117**(5): p. 374-380.
 23. Warnakulasuriya, S., *Oral potentially malignant disorders: A comprehensive review on clinical aspects and management*. Oral Oncol, 2020. **102**: p. 104550.
 24. Wils, L.J., et al., *Elucidating the Genetic Landscape of Oral Leukoplakia to Predict Malignant Transformation*. Clin Cancer Res, 2023. **29**(3): p. 602-613.
 25. Hanna, G.J., et al., *Nivolumab for Patients With High-Risk Oral Leukoplakia: A Nonrandomized Controlled Trial*. JAMA Oncol, 2024. **10**(1): p. 32-41.
 26. Kerr, A.R. and G. Lodi, *Management of oral potentially malignant disorders*. Oral Dis, 2021. **27**(8): p. 2008-2025.
 27. Zhang, M., et al., *Digital Image Analysis of Ki67 Heterogeneity Improves the Diagnosis and Prognosis of Gastroenteropancreatic Neuroendocrine Neoplasms*. Mod Pathol, 2023. **36**(1): p. 100017.
 28. Lin, L., et al., *Multifunctional photodynamic/photothermal nano-agents for the treatment of oral leukoplakia*. J Nanobiotechnology, 2022. **20**(1): p. 106.
 29. ., R.b.I.J.G.N.H.J.H.t.J.L.U.M.M.S.S.P., *Oral potentially malignant disorders and oral epithelial dysplasia*. 2017.
 30. Bhattarai, B.P., et al., *Recurrence in Oral Leukoplakia: A Systematic Review and Meta-analysis*. J Dent Res, 2024. **103**(11): p. 1066-1075.
 31. Li, X., et al., *Single-cell transcriptomic analyses reveal heterogeneity and key*

- subsets associated with survival and response to PD-1 blockade in cervical squamous cell carcinoma. Cancer Cell Int, 2025. 25(1): p. 90.*
32. Wang, P., et al., *Single-cell RNA sequencing unveils tumor heterogeneity and immune microenvironment between subungual and plantar melanoma. Scientific Reports, 2024. 14(1): p. 7039.*
33. Wang, Z., et al., *Comparative immunological landscape between pre- and early-stage LUAD manifested as ground-glass nodules revealed by scRNA and scTCR integrated analysis. Cell Commun Signal, 2023. 21(1): p. 325.*

2nd May 2025

Dear Dr. Li,

Thank you for the submission of your revised manuscript to EMBO Molecular Medicine. I am pleased to inform you that we will be able to accept your manuscript pending the following final amendments:

- 1) Authors: Please provide institutional email address for the co-corresponding author Xin Zeng.
- 2) Figures: Please rename supplementary figures to Figure EV1 etc. and update their callouts in the main manuscript file. EV figure legends should be renamed to "Expanded View Figure Legends".
- 3) Source data: Please provide source data for Figure S6. Remove folders for Fig 1C and F as these are deposited in external repository.
- 4) Please address all comments suggested by our data editors listed below:
 - o Figure legends:
 1. Please note that the exact p values are not provided in the legends of figures 2B, D, F, G, H, I, J, K; 3B, 4B, D, E, F, G; 5A, B, F, G, H, K, L; 6D, G, I; S2 B-D; S3 B-D; S4 A, S5 G-I; S6 B.
 2. Please indicate the statistical test used for data analysis in the legends of figures 2H, J; S5 E.
 3. Please note that the box plots need to be defined in terms of minima, maxima, centre, bounds of box and whiskers, and percentile in the legends of figures 2D, 5A.
 4. Please note that information related to n is missing in the legends of figures 2D, 3B, D-F; 4B, D, E, F, G; 5A, B, F, G, H, K, L; S2 B, C; S3 B, C, D; S4 A; S5 G-I.
 - Correct order of manuscript sections: Abstract / Keywords / The Paper Explained / Introduction / Results / Discussion / Methods / Data Availability / Acknowledgements / Disclosure and Competing Interests Statement / References / Main Figure Legends / Tables / Expanded View Figure Legends.
 - Limit keywords to max. 5.
 - Figure callouts should be in sequential order. Currently Fig 3A is called out before Fig 1. Please correct.
 - In Methods, provide the statement that informed consent was obtained from all human subjects and confirm that the experiments conformed to the principles set out in the WMA Declaration of Helsinki and the Department of Health and Human Services Belmont Report.
 - In Methods, add a paragraph describing immunoblotting and provide the antibody dilutions that were used for each antibody.
 - Indicate in legends exact n and exact p values, not a range, along with the statistical test used. To keep the figures "clear" some authors found providing an Appendix table Sx with all exact p-values preferable. You are welcome to do this if you want to.
 - Please remove Reagents and Tools Table and uploaded it as a separate file. Structured Methods section includes Reagents and Tools Table followed by a Methods and Protocols section. More information on how to adhere to this format as well as downloadable templates (.docx) for the Reagents and Tools Table can be found in our author guidelines: <https://www.embopress.org/page/journal/17574684/authorguide#structuredmethods>
 - An example of a paper with Structured Methods can be found here: <https://www.embopress.org/doi/full/10.1038/s44320-024-00037-6#sec-4>
 - Please remove "Supplementary Material".
 - Rename "Conflict of Interest statement" to "Disclosure Statement & Competing Interests" and place it after the "Acknowledgements". We updated our journal's competing interests policy in January 2022 and request authors to consider both actual and perceived competing interests. Please review the policy <https://www.embopress.org/competing-interests> and update your competing interests if necessary.
 - Please use the following format to report the accession number of your data:

[data type]: [full name of the resource] [accession number/identifier] ([doi or URL or identifiers.org/DATABASE:ACCESSION])

Please check "Author Guidelines" for more information.

<https://www.embopress.org/page/journal/17574684/authorguide#availabilityofpublishedmaterial>

- 1) Please correct the reference citation in the text and reference list. In the text a reference should be cited by author and year of publication. Include a space between a word and the opening parenthesis of the reference that follows. In the reference list, citations should be listed in alphabetical order. Where there are more than 10 authors on a paper, 10 will be listed, followed by "et al.". Also, please remove DOIs. Please check "Author Guidelines" for more information. <https://www.embopress.org/page/journal/17574684/authorguide#referencesformat>
- 5) Funding: Please merge it with "Acknowledgements".
- 6) Appendix: Please rename Suppl. Tables to "Appendix" and add a table of content with page numbers on the title page. Rename the tables to "Appendix Table S1" etc. and updated their callouts in the main manuscript file.
- 7) The Paper Explained: Please add it to the main manuscript text.
- 8) Synopsis:
 - Synopsis image: Please resize the image to 550 px-wide x 200-600 pixels high and upload it as a high-resolution jpeg file.
 - Please check your synopsis text and image before submission with your revised manuscript. Please be aware that in the proof

stage minor corrections only are allowed (e.g., typos).

9) As part of the EMBO Publications transparent editorial process initiative (see our Editorial at <http://embomolmed.embopress.org/content/2/9/329>), EMBO Molecular Medicine will publish online a Review Process File (RPF) to accompany accepted manuscripts. This file will be published in conjunction with your paper and will include the anonymous referee reports, your point-by-point response and all pertinent correspondence relating to the manuscript. Let us know whether you agree with the publication of the RPF and as here, if you want to remove or not any figures from it prior to publication. Please note that the Authors checklist will be published at the end of the RPF.

10) Please provide a point-by-point letter INCLUDING my comments as well as the reviewer's reports and your detailed responses (as Word file).

I look forward to reading a new revised version of your manuscript as soon as possible.

Yours sincerely,

Zeljko Durdevic

*** Instructions to submit your revised manuscript ***

- 1) a .docx formatted version of the manuscript text (including Figure legends and tables)
- 2) Separate figure files*
- 3) supplemental information as Expanded View and/or Appendix. Please carefully check the authors guidelines for formatting Expanded view and Appendix figures and tables at <https://www.embopress.org/page/journal/17574684/authorguide#expandedview>
- 4) a letter INCLUDING the reviewer's reports and your detailed responses to their comments (as Word file).
- 5) The paper explained: EMBO Molecular Medicine articles are accompanied by a summary of the articles to emphasize the major findings in the paper and their medical implications for the non-specialist reader. Please provide a draft summary of your article highlighting
 - the medical issue you are addressing,
 - the results obtained and
 - their clinical impact.This may be edited to ensure that readers understand the significance and context of the research. Please refer to any of our published articles for an example.
- 6) Author contributions: the contribution of every author must be detailed in a separate section.

7) EMBO Molecular Medicine now requires a complete author checklist (<https://www.embopress.org/page/journal/17574684/authorguide>) to be submitted with all revised manuscripts. Please use the checklist as guideline for the sort of information we need WITHIN the manuscript. The checklist should only be filled with page numbers where the information can be found. This is particularly important for animal reporting, antibody dilutions (missing) and exact values and n that should be indicated instead of a range.

8) Every published paper now includes a 'Synopsis' to further enhance discoverability. Synopses are displayed on the journal webpage and are freely accessible to all readers. They include a short stand first (maximum of 300 characters, including space) as well as 2-5 one sentence bullet points that summarise the paper. Please write the bullet points to summarise the key NEW findings. They should be designed to be complementary to the abstract - i.e. not repeat the same text. We encourage inclusion of key acronyms and quantitative information (maximum of 30 words / bullet point). Please use the passive voice. Please attach these in a separate file or send them by email, we will incorporate them accordingly.

You are also welcome to suggest a striking image or visual abstract to illustrate your article. If you do please provide a jpeg file 550 px-wide x 300-600px high.

9) A Conflict of Interest statement should be provided in the main text

10) Please note that we now mandate that all corresponding authors list an ORCID digital identifier. This takes <90 seconds to complete. We encourage all authors to supply an ORCID identifier, which will be linked to their name for unambiguous name identification.

Currently, our records indicate that the ORCID for your account is 0000-0001-5173-0781.

Link Not Available

11) Include a Reagents and Tools Table as part of the Methods section, which can be downloaded from our author guidelines (<https://www.embopress.org/page/journal/17574684/authorguide#structuredmethods>)

Photos 400-800 DPI

*Additional important information regarding figures and illustrations can be found at <https://bit.ly/EMBOPressFigurePreparationGuideline>. See also figure legend preparation guidelines: <https://www.embopress.org/page/journal/17574684/authorguide#figureformat>

***** Reviewer's comments *****

Referee #1 (Comments on Novelty/Model System for Author):

The study aim and design are really innovative

Referee #1 (Remarks for Author):

In this revised version of the manuscript, the authors have well addressed all issues raised by the reviewer. The manuscript can be acceptable for publication in this current form.

Referee #2 (Comments on Novelty/Model System for Author):

This is good, model improvement not necessary for this study

Referee #2 (Remarks for Author):

The review has been comprehensively performed

The authors addressed the remaining editorial issues.

16th May 2025

Dear Dr. Li,

We are pleased to inform you that your manuscript is accepted for publication and is now being sent to our publisher to be included in the next available issue of EMBO Molecular Medicine.

Zeljko Durdevic
Senior Editor
EMBO Molecular Medicine
